# Highly efficient intercellular spreading of protein misfolding mediated by viral ligand-receptor interactions

Shu Liu[1,10], André Hossinger[1], Stefanie-Elisabeth Heumüller[1], Annika Hornberger[1], Oleksandra Buravlova[1], Katerina Konstantoulea [2,3], Stephan A. Müller [4,5], Lydia Paulsen[1], Frederic Rousseau[2,3], Joost Schymkowitz[2,3], Stefan F. Lichtenthaler[4,5,6], Manuela Neumann[7,8], Philip Denner[1] & Ina M. Vorberg [1,9 ✉]

Protein aggregates associated with neurodegenerative diseases have the ability to transmit to unaffected cells, thereby templating their own aberrant conformation onto soluble homotypic proteins. Proteopathic seeds can be released into the extracellular space, secreted in association with extracellular vesicles (EV) or exchanged by direct cell-to-cell contact. The extent to which each of these pathways contribute to the prion-like spreading of protein misfolding is unclear. Exchange of cellular cargo by both direct cell contact or via EV depends on receptor-ligand interactions. We hypothesized that enabling these interactions through viral ligands enhances intercellular proteopathic seed transmission. Using different cellular models propagating prions or pathogenic Tau aggregates, we demonstrate that vesicular stomatitis virus glycoprotein and SARS-CoV-2 spike S increase aggregate induction by cell contact or ligand-decorated EV. Thus, receptor-ligand interactions are important determinants of intercellular aggregate dissemination. Our data raise the possibility that viral infections contribute to proteopathic seed spreading by facilitating intercellular cargo transfer.

[1] German Center for Neurodegenerative Diseases Bonn (DZNE), Venusberg Campus 1/ 99, 53127 Bonn, Germany. [2] VIB Center for Brain and Disease Research, Leuven, Belgium. [3] Switch Laboratory, Department of Cellular and Molecular Medicine, KU Leuven, Leuven, Belgium. [4] German Center for Neurodegenerative Diseases (DZNE), Munich, Germany. [5] Neuroproteomics, School of Medicine, Klinikum rechts der Isar, Technical University of Munich, 81675 Munich, Germany. [6] Munich Cluster for Systems Neurology (SyNergy), Munich, Germany. [7] Department of Neuropathology, University Hospital Tübingen, Tübingen, Germany. [8] Molecular Neuropathology of Neurodegenerative Diseases, German Center for Neurodegenerative Diseases (DZNE), Tübingen, Germany. [9] Rheinische Friedrich-Wilhelms-Universität Bonn, Venusberg Campus 1, 53127 Bonn, Germany. [10]Present address: German Federal Institute for Risk Assessment (BfR), German Centre for the Protection of Laboratory Animals (Bf3R), Max-Dohrn-Straße 8-10, 10589 Berlin, Germany. ✉email: ina.vorberg@dzne.de

Aberrant folding and aggregation of host-encoded proteins into ordered assemblies is a pathological hallmark of neurodegenerative diseases (ND), such as prion diseases, Alzheimer's disease (AD) and Parkinson's disease (PD). Disease-associated protein deposition usually starts locally, subsequently spreading stereotypically to other brain regions[1]. AD, the most prevalent neurodegenerative disease, is associated with the extracellular accumulation of Aβ amyloid and intracellular inclusion of misfolded microtubule-binding protein Tau as in neurofibrillary tangles[2,3]. The formation of Tau aggregates is also a hallmark of other ND, collectively known as tauopathies[4]. Transmissible spongiform encephalopathies or prion diseases constitute a special group of ND affecting humans and other mammals[5]. Prion diseases can occur sporadically, be caused by mutations in the prion protein gene or can be acquired by infection or iatrogenic transmission[6].

Pathogenic protein aggregates form in a time-limiting nucleation-dependent process in which soluble aggregation-prone proteins form oligomers that grow into highly ordered, beta-sheet-rich fibrils[7]. In vitro, the lag phase required for seed formation drastically shortens in the presence of pre-formed polymers that act as seeds[8]. Proteopathic seeds not only recruit monomeric protein in the affected cell but also in unaffected cells upon intercellular transmission, a process that likely underlies the often observed stereotypical spreading of protein misfolding in ND[1]. Intercellular transmission of proteopathic seeds is regarded as a common mechanism of ND[9]. The precise mechanism of intercellular aggregate transfer and induction of new aggregates is unclear but appears to involve release of free protein aggregates, direct cell-to-cell contact by cytonemes, such as tunneling nanotubes (TNTs)[10–12], or EV[13]. The extent to which these three routes contribute to the spreading of protein misfolding remains unclear.

EV are nanosized communication vesicles secreted under physiological and pathological conditions[14]. EV differ in their cellular origin, with some budding directly from the cell surface (so-called microvesicles), and others being secreted when multi-vesicular bodies of endosomal origin fuse with the plasma membrane and release exosomes into the extracellular space. Secreted vesicles are now generally referred to as EV due to the substantial overlap of microvesicles and exosomes in terms of size, surface markers, and function[15]. Soluble and aggregated proteins associated with a diverse number of ND have been found secreted by neurons and other cells either as free proteins or in association with EV[16–20]. While EV containing proteopathic seeds induce protein aggregation in vitro and in vivo, the efficiency of protein aggregate transfer and subsequent seeding through this route is unclear. Only a small fraction of released soluble or aggregated proteins are associated with EV, while the vast majority is freely secreted. For example, less than 1% of total secreted Aβ is associated with EV[16], and only 3% of total secreted α-synuclein was found in the EV fraction[21]. Rat cortical neurons have been shown to secrete Tau, but again very little (3 %) was associated with EV[18]. EV fractions purified from N2a cells with an aggregated Tau mutant induced Tau aggregation in less than 0,1 % of recipient cells, arguing that EV-mediated aggregate induction can be rather inefficient[18].

EV isolated from different donor cells exhibit marked cell tropism[22,23]. For cytosolic cargo release, EV usually merge with cellular membranes. EV docking and subsequent uptake are selective processes that require specific membrane interactions[23]. Consequently, receptor-ligand interactions between EV and recipient cells will likely modulate the spreading behavior of proteopathic seed cargo. While integrins and proteoglycans have been identified that adhere EV to target cells, most receptor-ligand pairs that underlie these targeted interactions are so far unknown[23,24].

We reasoned that the poor aggregate-inducing activity of some seed-containing EV could be due to the lack of specific ligands required for host receptor interactions and/or fusion. Membrane contact and fusion can be facilitated by viral glycoproteins that allow viruses to adhere to and penetrate their target cells. The contact with receptors leads to conformational transitions in the viral glycoproteins, thereby bringing the two membranes in close proximity and enforcing bilayer merger. Viral glycoproteins are routinely used to pseudotype genetically engineered viral vectors for efficient cargo delivery.

The vesicular stomatitis virus glycoprotein VSV-G is the sole surface protein of this virus belonging to the rhabdovirus genus that mediates the binding to the LDL receptor or its family members[25,26]. Ectopically expressed VSV-G is not only found on the cell surface but also decorates EV in the absence of other viral constituents[27]. It has been recently demonstrated that VSV-G enhances EV-mediated cargo delivery to recipient cells[27,28]. We here expressed VSV-G in cellular models that propagate different protein aggregates. Coculture of VSV-G-expressing donor cells with recipient cells strongly increased protein aggregate induction in the latter. Further, the expression of VSV-G also promoted the secretion of VSV-G-coated EV with enhanced aggregate-inducing capacity in recipient cells. Interactions between SARS-CoV-2 spike S protein and its receptor ACE2 similarly contributed to the spreading of cytosolic prions and Tau aggregates. Thus, efficient intercellular proteopathic seed transfer can be strongly controlled by receptor-ligand interactions. Further, our data raise the possibility that viral glycoproteins, expressed during acute or chronic infection, could facilitate the spreading of protein misfolding in vivo.

## Results

**Expression of viral glycoprotein VSV-G drastically increases cell-to-cell spreading of cytosolic prions.** To study the intercellular dissemination and propagation of proteinaceous seeds, we have implemented cellular models that rely on the ectopic expression of a yeast prion domain in the cytosol of mammalian cells. The classification of the *Saccharomyces cerevisiae* Sup35 translation termination factor as a prion protein is based on the fact that it can exist in a functional soluble isoform and as a cross-beta-sheet polymer that self-replicates by imprinting its conformation onto a soluble protein of the same kind[29]. The prion domain NM of Sup35 confers aggregation capacity and is separable from the translation termination activity of the carboxyterminal domain[30]. Sup35 NM can serve as a model protein to study inducible protein aggregation[31]. When expressed in the mammalian cytosol, the NM prion domain exists in a soluble state. Exposure of cells to highly ordered protein fibrils composed of recombinant NM leads to the aggregation of expressed NM, thereby inducing self-replicating NM prions that are heritable by progeny[31]. NM prions also transmit to naïve bystander cells by direct cell-to-cell contact[32] and EV[33]. The presence of NM aggregates can be monitored using automated confocal microscopy and image analysis (Supplementary Fig. 1).

Interestingly, we found that donor cell populations with aggregates drastically differ in their NM aggregate-inducing activity, an effect that also depended on the recipient cell line[33,34]. We hypothesized that one reason for the poor NM aggregate induction could be inefficient membrane contact and fusion of either EV with target cells or between the donor and recipient cells. Since vesicular stomatitis virus glycoprotein VSV-G binds to the broadly expressed LDL receptor family and increases cell-to-cell membrane contact[25], we assessed whether its expression facilitates protein aggregate transmission. Mouse neuroblastoma cells or human HEK cells engineered to express soluble NM-GFP

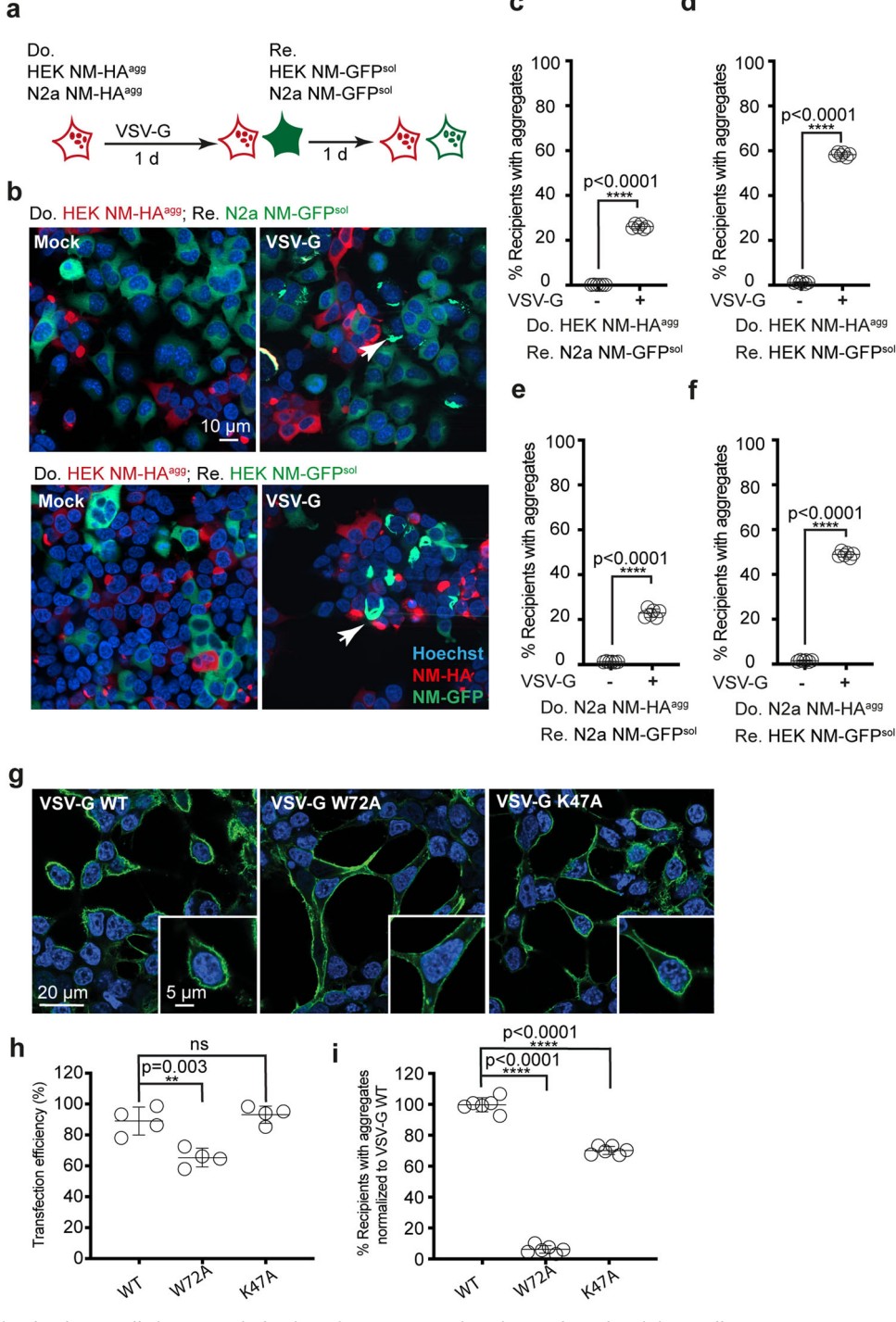

**Fig. 1 VSV-G expression by donor cells increases induction of Sup35 NM prions in cocultured recipient cells. a** A HEK NM-HA$^{agg}$ cell clone and a N2a NM-HA$^{agg}$ clone (red) with poor aggregate-inducing activity in recipients were chosen as donors (Do.). Donor cells were transiently transfected with plasmid coding for VSV-G or Mock transfected and cocultured with recipient cells expressing NM-GFP$^{sol}$ (Re.). **b** Transfected donor cells cocultured with recipient cells. Do Donor Re Recipient. NM-HA was stained using anti-HA antibodies. **c–f** Quantitative analysis of the percentage of recipient cells with induced NM-GFP aggregates (NM-GFP$^{agg}$). **g** Cell surface expression of VSV-G in donor HEK NM-HA$^{agg}$ cells. Cells were fixed 24 h post transfection (VSV-G antibody 8GF11). Insets show individual cells. **h**. Transfection efficiency in HEK NM-HA$^{agg}$ cells. Shown is the percentage of cells expressing VSV-G. **i**. VSV-G (WT) or VSV-G mutant transfected donor cells were cocultured with recipient HEK NM-GFP$^{sol}$ cells. NM-GFP aggregate induction was determined 1-day post coculture. Induction rates related to WT VSV-G expression were set to 100 %. All data are shown as the means ± SD from four (**h**) or six (**c–f**, **i**) replicate cell cultures. Three (**c–f**, **h**, **i**) independent experiments were carried out with similar results. *P* values were calculated by two-tailed unpaired Student´s *t* test (**c–f**) or one-way ANOVA (**h**, **i**). Source data are provided as a Source Data file.

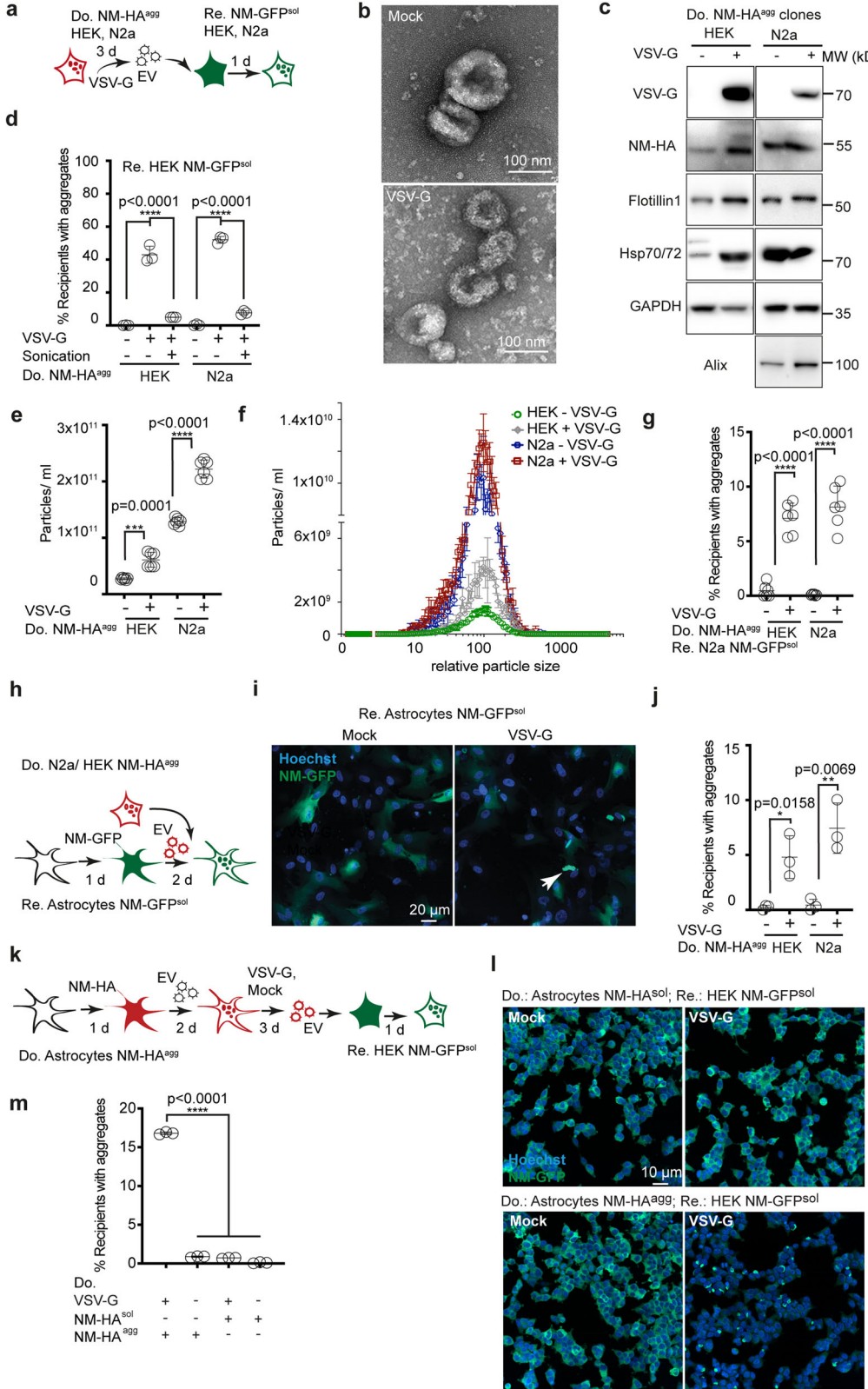

(NM-GFP^sol) (Supplementary Fig. 2a) and endogenously expressing LDL receptor (Supplementary Fig. 2b) served as recipients. As aggregate-bearing donors, we chose HEK NM-HA^agg cells[34] and N2a NM-HA^agg clone 2E[32] due to their low NM aggregate induction rates when cocultured with recipient cells. As controls, HEK or N2a cells expressing soluble NM-HA (NM-HA^sol) served as donors. When aggregate-bearing donor cells transfected with

empty or VSV-G coding plasmid were cocultured with N2a or HEK NM-GFP^sol recipients (Fig. 1a, b, Supplementary Fig. 2c), VSV-G drastically increased the percentage of recipient cells with induced NM-GFP aggregates (Fig. 1c–f). No NM-GFP aggregate formation was observed when donor cells expressed only soluble NM-HA (Supplementary Fig. 2d, e). In several instances, we observed syncytia formation, likely due to the fusogenic activity of

**Fig. 2 Pseudotyping EV with VSV-G drastically increases their intercellular NM aggregate induction efficiency. a** HEK or N2a NM-HA$^{agg}$ donors (Do.) were transfected with VSV-G coding plasmid or empty vector. Recipient (Re.) HEK or N2a cells expressing NM-GFP$^{sol}$ were exposed to EV. **b** Transmission electron microscopy images of EV isolated from transfected HEK NM-HA$^{agg}$ donor cells. **c** Western blot of EV lysates. **d** Percentage of recipient HEK NM-GFP cells with induced NM-GFP aggregates upon exposure to EV. Sonication (5 min, 100 % power) was used to destroy EV. **e**. Particle numbers of EV from VSV-G transfected or Mock transfected donors determined 3-day post transfection. **f** Size distribution of different EV. **g**. EV preparations adjusted to comparable EV numbers were added to recipient N2a NM-GFP$^{sol}$ cells. Recipient cells were analyzed for the percentage of cells with induced NM-GFP aggregates. **h** Human astrocytes expressing NM-GFP$^{sol}$ were exposed to VSV-G-pseudotyped or non-pseudotyped EV. NM-GFP aggregate induction was monitored 2-day post EV exposure. **i**. Astrocytes exposed to EV from HEK cells harboring NM-HA$^{agg}$. Arrowhead depicts aggregated NM-GFP. **j** Percentage of astrocytes with NM-GFP$^{agg}$ following exposure to EV from transfected donors. **k** Astrocytes were transduced with lentivirus coding for NM-HA. Cells were subsequently exposed to VSV-G-pseudotyped EV from HEK NM-HA$^{agg}$ cells to generate a donor population with a high percentage of NM-HA$^{agg}$ astrocytes. Donors were transduced with Mock lentivirus or VSV-G-coding lentivirus. EV harvested after medium exchange and adjusted to comparable particle numbers were added to HEK cells expressing NM-GFP$^{sol}$. **l** HEK NM-GFP$^{sol}$ cells 24 h postexposure to VSV-G or Mock EV harvested from astrocytes. **m**. Percentage of HEK cells with induced NM-GFP aggregates upon exposure to astrocyte EV. All data are shown as the means ± SD from three (**d**, **j**, **m**) or six (**e**, **g**) replicate cell cultures. Three (**d**, **e**, **g**, **j**) independent experiments were carried out with similar results. Experiment (**m**) was carried out once with three independent cell cultures per group. $P$ values were calculated by two-tailed unpaired Student´s $t$ test (**e**, **g**, **j**) or one-way ANOVA (**d**, **m**). Source data are provided as a Source Data file.

VSV-G (Supplementary Fig. 2d). Aggregates costained for both the NM-GFP and NM-HA, often associated with syncytia, were present in cocultures to different degrees. In cocultures with N2a donors, approximately 26.2 ± 5.3% aggregate-bearing recipient N2a or 14.4 ± 2.9% HEK exhibited costained aggregates. When HEK cells served as donors, costained aggregates were observed in 50 ± 3.2% (N2a) or 4.0 ± 0.7% (HEK) aggregate-bearing recipients (Supplementary Fig. 2f). The prion-inducing activity of donor cells expressing VSV-G strongly depended on the fusogenic activity of the viral protein, as nonfusogenic VSV-G mutant W72A[35,36] was unable to promote NM-GFP aggregation (Fig. 1g–i). Reduced aggregate induction was observed when donors expressed VSV-G mutant K47A that exhibits fusion activity but lacks LDL receptor family recognition (Fig. 1g–i)[37], suggesting that at least some intercellular membrane fusion occurred independently of binding to LDL receptor family members[37].

**VSV-G mediates efficient vesicular dissemination of cytosolic NM prions**. VSV-G is present in an inactive prefusion state at neutral pH that becomes activated by low pH in the endolysosomal system[38]. The occasional syncytia formation argued that some of the aggregate induction events could be due to extracellular VSV-G activation and subsequent cell fusion rather than contact and fusion at local cell contacts, such as cytonemes. We tested the effect of VSV-G pseudotyped EV on NM-GFP aggregate induction (Fig. 2a). Expression of VSV-G proved non-toxic to donor cells in our experimental setup (Supplementary Fig. 3a). EV fractions isolated from VSV-G expressing donors contained a heterogeneous population of vesicles (Fig. 2b) and stained positive for VSV-G, NM-HA, and EV marker proteins Flotillin1, Hsp70/72, Alix, and GAPDH (Fig. 2c). The presence of VSV-G on EV strongly increased NM-GFP aggregation when added to recipient cells (Fig. 2d, Supplementary Fig. 3b). Destruction of EV by sonication strongly reduced aggregate induction (Fig. 2d). Direct comparison of the effect of sonication on EV or recombinant NM fibrils confirmed that sonication abolished the seeding activity of VSV-G-coated EV, while increasing the seeding activity of fibrils (Supplementary Fig. 3c–d). In line with our previous findings[33], the aggregation state of EV-associated NM-HA remained relatively unaffected by sonication (Supplementary Fig. 3e). Providing VSV-G in trans only slightly increased aggregate-inducing activity, arguing that VSV-G needs to be present on NM-HA aggregate containing EV to increase seeding (Supplementary Fig. 3f, g). We conclude that intact EV decorated with VSV-G were required for efficient prion spreading.

The effect of VSV-G expression on prion induction could either be due to enhanced production of EV, enhanced membrane docking and fusion of EV and recipient cells, or both. Consistent with previous findings[28], we observed an approximately two-fold increase in EV release upon transient expression of VSV-G in donor cell clones (Fig. 2e). VSV-G pseudotyping did not change the size distribution of released EV (Fig. 2f). When adjusted for particle numbers, still an increase in intercellular aggregate induction was observed when EV were pseudotyped with VSV-G (Fig. 2g). EV pseudotyped with nonfusogenic VSV-G variant W72A exhibited very low aggregate-inducing activity (Supplementary Fig. 3h, i). Mutant K47A with impaired binding to LDL receptor family members still resulted in aggregate induction, albeit to a lower extent (Supplementary Fig. 3h, i). This suggests that VSV-G decorated EV might be able to use alternative receptors and uptake pathways, in line with multiple entry routes used by EV[39].

We assessed if viral glycoproteins also enhance intercellular protein aggregate induction in primary cells. Protein aggregation was successfully induced in primary human astrocytes expressing NM-GFP using VSV-G-coated EV from HEK NM-HA$^{agg}$ donors (Fig. 2h–j). Further, VSV-G expression strongly increased intercellular aggregate induction in cocultures when human primary astrocytes served as donors (Supplementary Fig. 4a–g). Moreover, EV isolated from astrocyte donors transduced with VSV-G (Fig. 2k–m) efficiently induced NM-GFP aggregation in recipients (Fig. 2l, m). We conclude that VSV-G expressed by donor cells can strongly increase intercellular NM prion induction by mediating efficient contact and fusion of apposing cell membranes, as well as of EV and target cells.

**VSV-G-pseudotyped EV preferably enter cells by clathrin-mediated endocytosis and fuse at low pH for endosomal escape**. Vesicular stomatitis virus and VSV-G pseudovirions infect cells by receptor-mediated endocytosis via clathrin-coated pits[40,41], with subsequent low pH triggered membrane fusion and viral genome escape from early endosomes[38]. Genetic and pharmacologic manipulations known to inhibit infection with VSV or VSV-G-pseudotyped lentivirus[40,42,43] were used to assess entry of VSV-G-coated EV. VSV-G-pseudotyped EV induced NM-GFP aggregates in recipient cells, as soon as 80 min post EV exposure (Fig. 3a, suppl. movie). Repression of clathrin heavy chain (CLTC) expression (Fig. 3b–d) resulted in clustering of VSV-G-coated EV on the cell surface (Fig. 3e) and decreased aggregate induction in recipients, albeit to different degrees depending on the time point of analysis post EV exposure (Fig. 3f–h). Dynasore[44], a drug preventing scission of clathrin-coated pits,

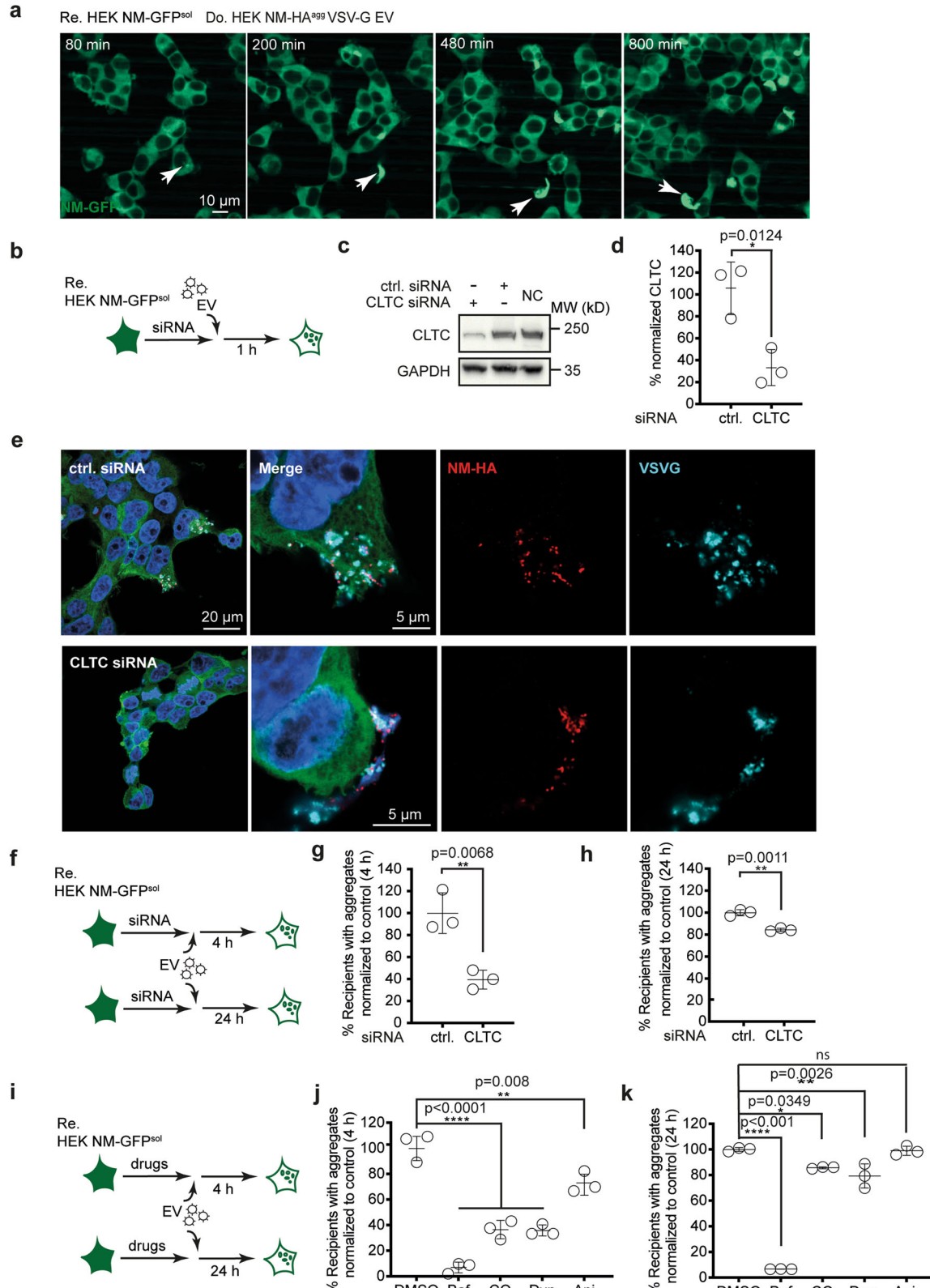

impaired aggregate induction (Fig. 3i–k). Raising the endosomal pH with Chloroquine, shown to partially inhibit VSV infection, had similar effects on aggregate induction by VSV-G-pseudotyped EV[45]. Aggregate induction was profoundly blocked in the presence of BafA1, an inhibitor of the $H^+$-ATPase required for endosomal acidification, a process that triggers VSV-G mediated fusion and infection[46]. By contrast, Apilimod, a pIKfyve inhibitor with no effect on VSV-G pseudovirion infection[46] had either only a small or no effect on aggregate induction. Thus, VSV-G coated EV preferentially enter cells by clathrin-mediated endocytosis, while fusion and endosomal escape are pH-dependent, in line with VSV infection[40,41]. As neither a VSV-G mutant unable to bind to its receptors nor inhibition of clathrin-dependent endocytosis completely blocked

**Fig. 3 VSV-G-pseudotyped EV are preferentially taken up by clathrin-mediated endocytosis. a** Kinetics of EV-mediated NM-GFP aggregate induction. HEK NM-GFP[sol] cells were exposed to EV from HEK NM-HA[agg] cells transfected with VSV-G plasmid. Life cell imaging was initiated immediately after EV exposure. Arrowhead depicts aggregate. **b** HEK NM-GFP[sol] cells were transfected with control siRNA or siRNA against clathrin heavy chain CLTC. 72 h later, cells were exposed to VSV-G-pseudotyped EV from HEK NM-HA[agg] donors. Cells were assessed for EV uptake 1 h post EV exposure. **c** Western blot of CLTC knock-down 72 h post transfection. **d** Quantitative analysis of CLTC knock-down. **e** Detection of VSV-G and NM-HA 1 h post EV addition. **f** Recipient HEK NM-GFP[sol] cells were transfected with CLTC or control siRNA and 72 h later exposed to EV for 4 or 24 h. **g, h** Percentages of CLTC siRNA transfected recipients with NM-GFP[agg] compared to induced recipient cells transfected with control siRNA set to 100 %. Cells were imaged 4 (**g**) and 24 h (**h**) post EV exposure. **i** Recipient cells were exposed to inhibitors for 1–2 h and aggregate induction was monitored either 4 or 24 h post EV exposure. Baf Bafilomycin, CQ Chloroquine, Dyn Dynasore, Api Apilimod. **j, k** Percentages of drug-treated recipients with NM-GFP[agg] compared to recipients treated with DMSO set to 100%. Recipient cells with NM-GFP[agg] were analyzed 4 h (**j**) or 24 h (**k**) post EV addition. All data are shown as the means ± SD from three (**d, g, h, j, k**) replicate cell cultures. Three independent experiments were carried out with similar results (**d**). EV experiments with CLTC knock-down or drug treatment were performed once at 4 h and once at 24 h (**g, h, j, k**). *P* values calculated by two-tailed unpaired Student´s *t* test (**d, g, h**) or one-way ANOVA (**j, k**). ns nonsignificant. Source data are provided as a Source Data file.

EV-mediated aggregate induction, VSV-G decorated EV likely also employ alternative entry routes.

**Enhanced intercellular transmission of Tau aggregation upon VSV-G expression.** Accumulating evidence suggests that seeding-competent Tau species can spread from cell-to-cell by EV[18]. To assess the effect of receptor-ligand interactions on intercellular Tau aggregate induction, we used a previously published Tau cell model[47]. HEK cells stably expressing the aggregation competent Tau repeat-domain (amino acid residues 244–372) with mutations P301L/V337M fused to GFP (Tau-GFP) were exposed to homogenates extracted from affected brain regions from patients with AD, cortical basal degeneration (CBD), progressive supra-nuclear palsy (PSP) or frontotemporal lobar degeneration with Tau pathology (FTLD-Tau). All patient brain homogenates contained aggregated Tau, as revealed by pronase digestion (Supplementary Fig. 5a). Upon limiting dilution cloning, we established HEK cell clones Tau-GFP[AD], Tau-GFP[FTLD], Tau-GFP[PSP], and Tau-GFP[CBD] stably producing Tau aggregates (Fig. 4a). Sedimentation assays and pronase treatment demonstrated the presence of aggregated Tau-GFP in all cell clones exposed to patient brain homogenate, but not in control cells exposed to control brain homogenate (Fig. 4b, Supplementary Fig. 5b). Donor cell clones propagating Tau aggregates exhibited aggregate-inducing activity in cocultured Tau-FusionRed (Tau-FR[sol]) recipients only when transfected with VSV-G (Fig. 4c–e). Likewise, no Tau-FR foci were detected when donors expressed only soluble Tau-GFP (Supplementary Fig. 5c–e). Few aggregates induced by HEK Tau-GFP[AD] cells were both GFP- and FR-positive (1.4 ± 0.1). Tau-FR aggregation was also induced in cocultured human primary astrocytes expressing soluble Tau-FR (Supplementary Fig. 5f, g).

We next tested the effect of VSV-G expression on EV-mediated Tau aggregation. VSV-G pseudotyped EV fractions from donor cells containing VSV-G, aggregated Tau-GFP and EV markers Flotillin1 and Hsp70/72 (Fig. 4f, g) exhibited a cup-shaped morphology by electron microscopy (Supplementary Fig. 5h). Addition of VSV-G-pseudotyped EV to recipient cells increased Tau aggregate induction (Fig. 4h–j, Supplementary Fig. 5i). Destruction of EV by sonication did not affect EV-associated Tau-GFP aggregates (Supplementary Fig. 5j), yet basically abolished aggregate induction (Fig. 4j). VSV-G expression also increased the number of particles released by donors with little effect on size distribution of EV (Fig. 4k, l). VSV-G pseudotyped EV adjusted to particle numbers comparable to control also resulted in increased Tau aggregate induction (Fig. 4m). We conclude that intact EV decorated with viral glycoprotein VSV-G efficiently transmit seeding-competent Tau.

**VSV-G-pseudotyped EV efficiently transmit scrapie prions to recipient cells.** Transmissible spongiform encephalopathy (TSE) agents are composed of misfolded prion protein PrP[5]. The conversion of cellular prion protein (PrP[C]), a protein tethered to the cell membrane by a glycosylphosphatidyl-anchor, into its infectious aggregated isoform PrP[Sc], occurs on the cell surface or along the endocytic pathway[48]. It has been shown that prion-infected N2a cells release prions associated with EV[19,49]. VSV-G-pseudotyped EV isolated from 22 L prion-infected N2a cells (N2a[22L]) successfully induced infection in permissive murine fibroblast cell line L929[50] and CAD5 cells[51] (Fig. 5a, b). VSV-G pseudotyping strongly affected PrP[Sc] accumulation in recipient cells (Fig. 5c) and also increased the number of cells containing PrP[Sc] aggregates (Fig. 5d–f). As observed before, VSV-G expression strongly increased particle release (Fig. 5g). When recipient cells were exposed to comparable numbers of EV, VSV-G expression resulted in increased infection of L929 cells (Fig. 5h, i). We conclude that the expression of viral ligand VSV-G drastically increases the capacity of donor cells to transmit both cytosolic and membrane-anchored proteopathic seeds to recipient cells.

**Increased proteopathic seed spreading upon SARS-CoV-2 spike S expression.** Next, we tested if glycoproteins associated with human pathogenic viruses could contribute to protein aggregate spreading. SARS-CoV-2 is a novel Betacorona virus that has become a pandemic threat with millions of confirmed cases since its outbreak in December 2019[52]. SARS-CoV-2 binds to its target cells by interaction of its spike protein S with the human angiotensin-converting enzyme 2 receptor (ACE2)[46,52,53]. Spike S is a transmembrane protein that is cleaved by host proteases into two subunits responsible for receptor binding and fusion with the host cell membrane. We assessed if ectopic expression of spike S by donor cells modulates proteopathic seed spreading in our models (Fig. 6a). Donor cell populations propagating NM-HA or Tau-GFP aggregates were transfected with a vector coding for SARS-CoV-2 spike S or Mock control vector. Both precursor protein and cleaved subunit S1 were identified in lysates of transfected cells, demonstrating that spike S was accurately processed by host proteases (Fig. 6b). HEK cells overexpressing ACE2 or Vero cells highly susceptible to SARS-CoV-2[52] were used as recipients (Fig. 6a). Coculture with aggregate-bearing HEK donors overexpressing spike S increased aggregate induction in recipients (Fig. 6c–e), an event clearly dependent on the expression of the viral ligand (Fig. 6f–h). Importantly, spike S also associated with the EV fraction secreted by donor cells (Fig. 6i). Isolated EV were also tested for their aggregate-inducing capacity in their respective target cells (Fig. 6j). Spike S expression did not affect EV secretion (Fig. 6k) but significantly increased numbers of recipient cells with

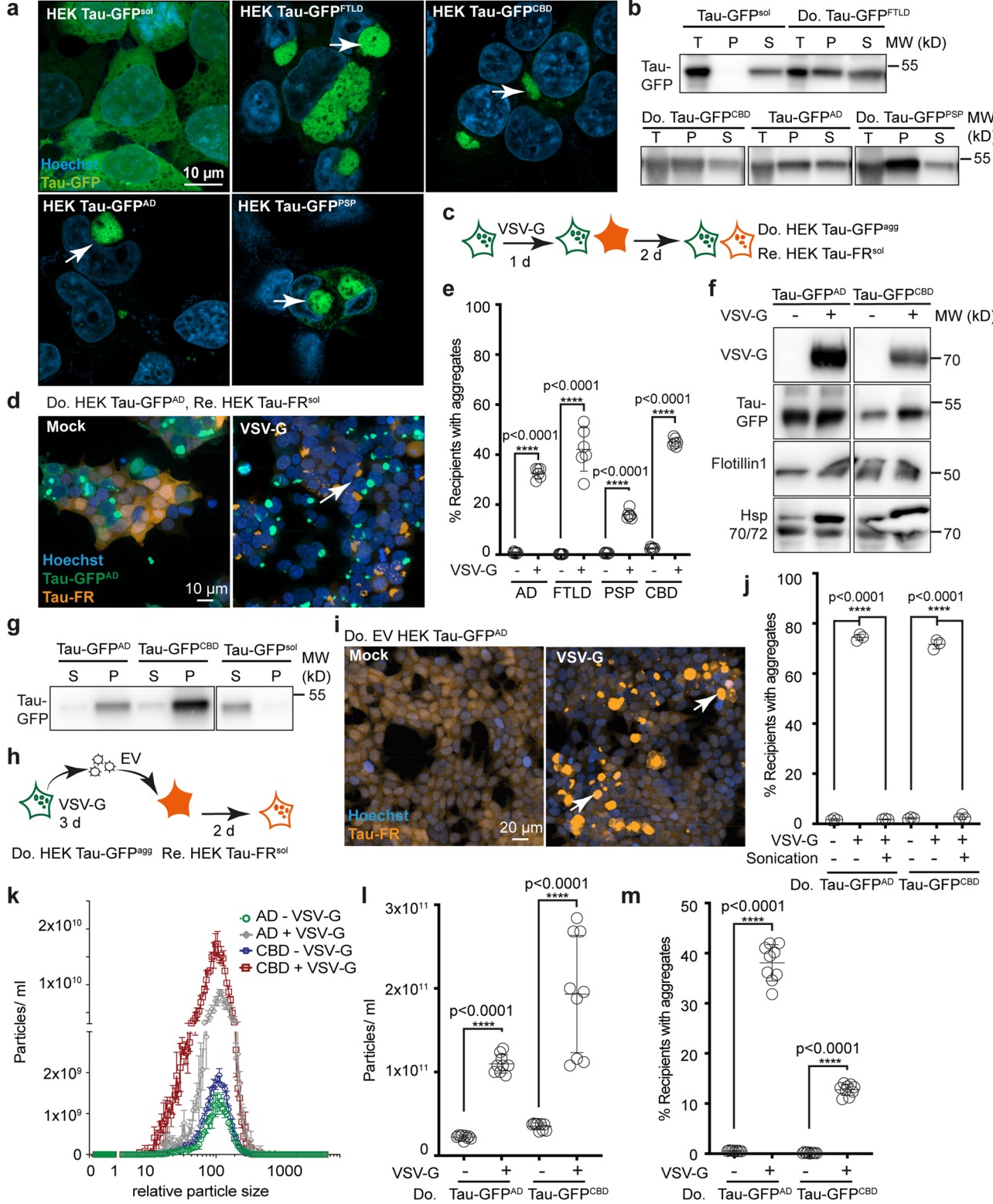

induced aggregates (Fig. 6l, m). We conclude that glycoproteins from different viral genera facilitate the spreading of proteopathic seeds, suggesting that viral glycoproteins could generally contribute to the intercellular exchange of cellular components.

## Discussion

Dissemination of protein aggregates between cells can occur by secretion of membrane-free naked aggregates, secretion as vesicular cargo, or via direct cell-to-cell contact. Recent findings that disease-associated proteins such as Tau and α-synculein are predominantly secreted in a nonmembrane bound state[16,18,21,54] have focused research on uptake mechanisms of free oligomers or fibrils, thereby uncovering heparan sulfate proteoglycans[54,55] and LRP1[56] as receptors. However, proteopathic seeds are also secreted by small exosome-like vesicles and large membrane-shed vesicles[18,20,57,58]. The extent to which EV contribute to seed

**Fig. 4 VSV-G expression enhances Tau aggregate induction. a** HEK Tau-GFP population and HEK clones propagating aggregated Tau-GFP. A HEK cell clone expressing soluble Tau-GFP (HEK Tau-GFP$^{sol}$) was exposed to 1% brain homogenates from tauopathy patients. Clones with visible Tau-GFP foci were isolated (Tau-GFP$^{AD}$, Tau-GFP$^{FTLD}$, Tau-GFP$^{PSP}$, Tau-GFP$^{CBD}$). Arrowheads: Tau-GFP foci. **b** Sedimentation assay of cell lysates from HEK Tau-GFP$^{sol}$ cells and clones propagating Tau-GFP$^{agg}$. T: total cell lysate; P: pellet; S: supernatant. **c** Transfected HEK Tau-GFP$^{agg}$ clones were cocultured with HEK cells stably expressing soluble Tau-FusionRed (HEK Tau-FR$^{sol}$). **d** Transfected HEK Tau-GFP$^{AD}$ cells were cocultured with recipient cells. Foci were monitored 2-day post coculture. Arrow: Tau-FR foci. **e** Percentage of cocultured recipient cells with induced Tau-FR foci. **f** Western blot of EV. **g** Sedimentation assay of Tau in EV (anti-Tau antibody ab64193). **h** EV from transfected donors were added to HEK Tau-FR$^{sol}$ cells. **i** Exposure of HEK Tau-FR$^{sol}$ cells to EV from transfected HEK Tau-GFP$^{AD}$. **j** Percentage of recipient cells with induced Tau-FR foci following EV addition. EV were also subjected to sonication (5 min, 100 % power). **k** Size distribution of EV. **l** EV released from transfected HEK Tau-GFP$^{AD}$ or Tau-GFP$^{CBD}$ donors were determined 3-day post transfection. **m** EV adjusted to comparable particle numbers were added to HEK Tau-FR$^{sol}$ cells. Cells with induced TauFR$^{agg}$ were determined 2 days later. All data are shown as the means ± SD from three (**j**), six (**e**), or nine (**l**, **m**) replicate cell cultures. Three (**e**, **j**, **l**, **m**) independent experiments were carried out with similar results. $P$ values calculated by two-tailed unpaired Student´s $t$ test (**e**, **l**, **m**) or one-way ANOVA (**j**). Source data are provided as a Source Data file.

transmission is unknown. We reasoned that proteopathic seed transmission involving direct membrane contacts between the donor and recipient cell, as well as of EV and target cells is at least partially controlled at cell entry. Here we demonstrate that the expression of two independent viral glycoproteins, which mediate receptor interaction and subsequent merger of opposing membranes, increased intercellular aggregate induction in cocultured cells and by EV. Our results demonstrate that the efficiency of docking and fusion of opposing membranes strongly influences intercellular aggregate induction. The fact that intercellular dissemination of three independent proteopathic seeds could be strongly increased by receptor-ligand interactions clearly shows that mechanisms of intercellular protein aggregate transfer are overlapping. Interestingly, VSV-G and spike S differed in their effect on intercellular aggregate induction. Of note, also titers of pseudotyped lentivirus are reduced up to 100-fold when VSV-G is replaced by spike S[59]. While glycoprotein expression increased intercellular aggregate induction in all cell lines and with all protein aggregates tested, the highest induction efficiency was not necessarily correlated with the highest glycoprotein expression levels. This suggests that other factors, such as cell (clonal) differences, transgene expression levels and/or protein aggregate conformation also influence intercellular aggregate transmission[32–34]. Our results further argue that, if equipped with suitable ligands for membrane interaction and fusion, EV represent highly effective vehicles for transfer of seeding-competent cargo. By contrast, insufficient receptor-ligand interactions constitute barriers to EV-mediated proteopathic seed spreading that can obscure the actual seeding capacity of packaged cargo.

The effect of viral glycoproteins on seed transmission suggests that certain viral infections could contribute to the dissemination of proteopathic seeds and ultimately modulate progression of protein misfolding diseases. Microbial brain infections have long been suspected to play a role in pathogenesis of ND[60]. Several neurotrophic viruses causing lifelong persistent infections, such as Herpesviridae, are upregulated in the CNS of ND patients and have been implicated in ND etiology[61,62]. Further, approximately 25% of HIV patients not undergoing combination antiretroviral therapy develop neurological disorders associated with diffuse Aβ plaque deposition and Tau neurofibrillary tangles[63]. Viral gene products can be directly neurotoxic[64,65] or indirectly elicit neuroinflammatory processes. Viral infections can also affect processing, deposition and clearance of ND-related proteins[66]. The results presented here suggest that viral glycoproteins could also contribute to the spreading and subsequent accumulation of disease-associated protein aggregates. In vitro evidence for the increase of protein aggregate spreading by viral infections comes from coinfection of fibroblasts with scrapie and Moloney leukemia retrovirus[67]. Increased prion

spreading was attributed to the expression of Gag capsid protein that accelerates secretion of prion-containing EV[67]. Gag expression also increased persistence of prion infection in an epithelial cell line infected with chronic wasting disease prions, demonstrating that some viral proteins alone are sufficient to modulate spreading of heterologous pathogens[68]. Of note, coinfection of mice with retrovirus and prions had no effect on disease incubation times, potentially because prime target cells of scrapie strains and γ-retroviruses differ[67,69].

We demonstrated that viral glycoproteins VSV-G and CoV-2 spike S alone are sufficient to increase intercellular spreading of protein aggregation. Interestingly, virally infected cells also secrete EV that promote viral spreading directly or indirectly[70,71]. For example, enveloped and nonenveloped viruses such as hepatitis A virus efficiently exploit EV for non-lytic cellular egress and subsequent infection[72]. EV decorated with viral glycoproteins, termed subviral particles (SVP), are released during infection by a diverse variety of viruses, including neurotrophic viruses such as HIV, Influenza, or Herpes viruses[71]. SVP lack capsid and viral genomes and often drastically outnumber viral particles. SVP contribute to viral immune evasion but could also increase intercellular dissemination of cargo due to efficient attachment and membrane fusion[73]. Viruses also spread by fusion of cellular plasma membranes, intercellular membranous connections such as cytonemes or TNTs, or virus-induced cell interfaces reminiscent of tight junctions or synapses[74]. For example, HIV glycoprotein and its receptor concentrate at filopodial tips of donor and recipient cells, thereby likely driving tip fusion and establishment of cytonemes for efficient cell-to-cell infection[75]. Viral infection through direct cell-to-cell contacts can be 2–3 orders of magnitude more efficient than by released viruses. Similar intercellular communication pathways are exploited by proteopathic seeds for dissemination. Fusion-competent viral glycoproteins, produced during viral infection, could thus strongly enhance seed transmission by these routes.

VSV is a rhabdovirus infecting ungulates that occasionally causes zoonotic flu-like infections in humans, and thus does not play a role in proteopathic seed spreading in ND. CoV-2 in humans primarily manifests as a respiratory illness, but neurological symptoms are present in 25% of acute cases and can be linked to direct involvement of the central nervous system[76]. Neuro-SARS symptoms comprise mostly nonspecific symptoms but rare cases of stroke, ataxia, seizures, or encephalitis have been reported[77]. Causes are likely multifactorial, including indirect mechanisms, such as systemic immune activation or neuroinflammation. Direct neuroinvasion has been demonstrated in autopsy cases, transgenic mice expressing human ACE2, and infected brain organoids[78–80]. Routes of neuroinvasion, as evident by the presence of SARS-CoV-2 viral RNA in the CNS[81,82], could be through the olfactory transmucosal route[83], vascular or other routes[84]. COVID-19

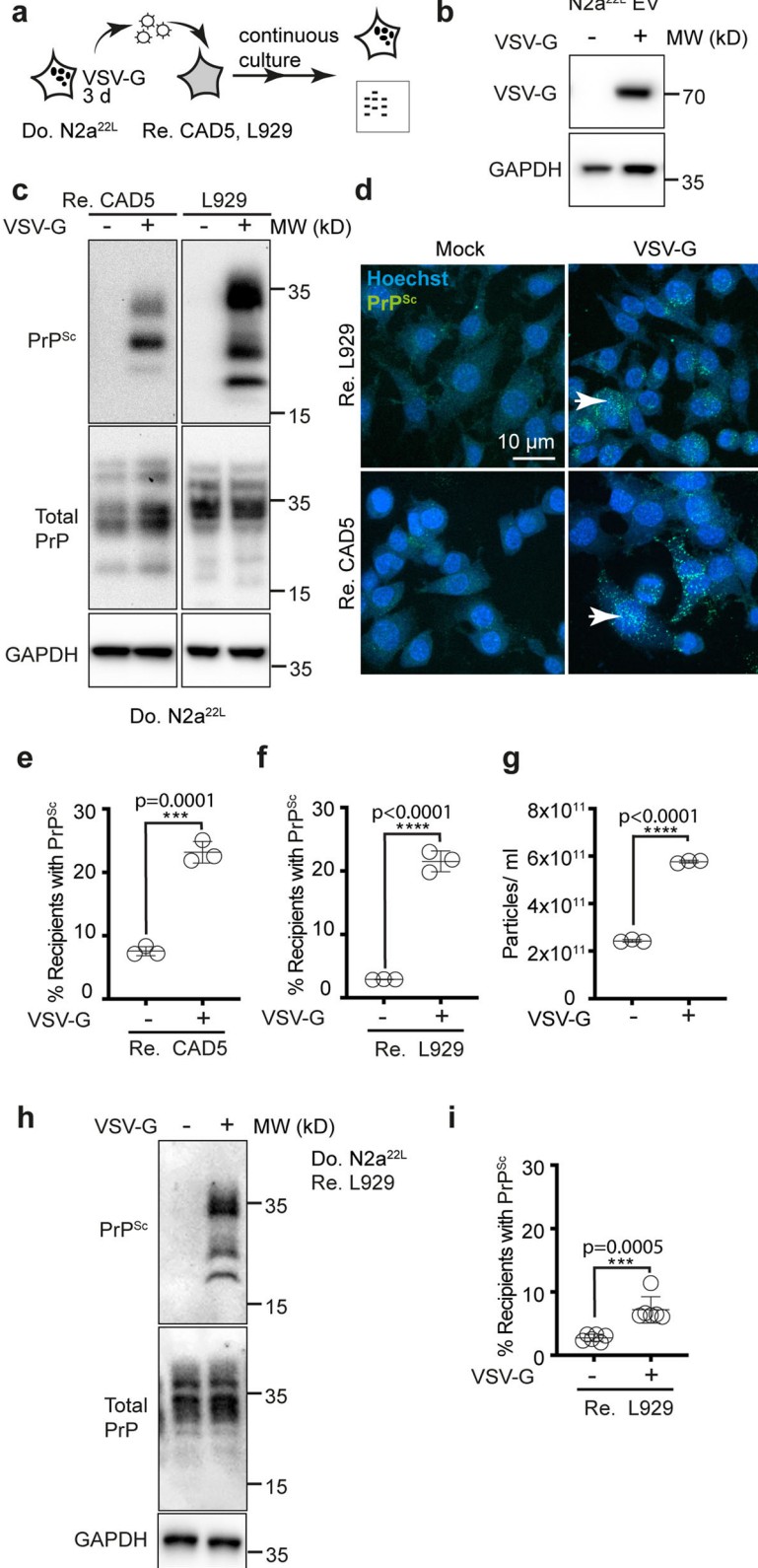

infections may increase the risk for developing neurological or ND later in life either directly or indirectly[76] but further research is required to clarify potential links. The results presented here argue that research should be intensified to clarify the effect of viruses on prion-like progression of protein aggregation in neurodegenerative and other protein misfolding diseases.

## Methods

**Human brain samples and ethics statement**. Research was performed in accordance with the Declaration of Helsinki. Frozen postmortem brain tissue samples from neuropathologically confirmed cases of AD, FTLD-tau, CBD, PSP, and control were provided by the Brain Bank associated with the University Hospital and DZNE Tübingen. In this Brain Bank, material and data are sampled and collected from donors upon written informed consent for brain

**Fig. 5 VSV-G increases spreading of transmissible spongiform encephalopathy agent. a** EV from transfected N2a cells persistently infected with scrapie strain 22 L (N2a$^{22L}$) were added to prion-permissive CAD5 or L929 cells. Recipient cells were passaged 7–8 times before PrP$^{Sc}$ formation was monitored. **b** Western blot of EV from donor cells. **c** Exposure of recipient cells to VSV-G-pseudotyped EV leads to infection. GAPDH and total PrP were detected on one blot (passage 7). Note that PrP$^C$ and PrP$^{Sc}$ run as unglycosylated, mono-glycosylated, and di-glycosylated bands. Total PrP refers to both the PrP$^C$ and PrP$^{Sc}$. Glycosylation profiles of PrP$^C$ and PrP$^{Sc}$ differ[90]. Nine times more PrP$^{Sc}$ sample was loaded. **d** Detection of PrP$^{Sc}$ in recipient cells exposed to EV from scrapie-infected donor cells eight passages postinfection. Arrowheads: PrP$^{Sc}$. **e**, **f** Quantitative analyses of recipients with PrP$^{Sc}$ puncta following exposure to EV from transfected donors. **g** Particles released from transfected N2a$^{22L}$ cells. **h** Western blot demonstrating PK-resistant PrP$^{Sc}$. EV were adjusted to comparable particles numbers before addition to cells. Cells were analyzed four passages postinfection. Note that by then EV have been diluted out $10^4$ times. Loading as above. **i** Quantitative analyses of the presence of PrP$^{Sc}$ in L929 cells exposed to VSV-G-decorated EV as in (**e**). Cells were analyzed 8 passages postinfection. All data are shown as the means ± SD from three (**e**–**g**) or six (**i**) replicate cell cultures. Three (**e**–**g**, **i**) independent experiments were carried out with similar results. *P* values calculated by two-tailed unpaired Student´s *t* test (**e**–**g**, **i**). Source data are provided as a Source Data file.

autopsy and the use of the material and clinical information for research purposes obtained by the probands or their legal representative according to the approval of the responsible ethic committee ("Ethik-Kommission, Medizinischen Fakultät der Eberhard-Karls-Universität und am Universitätsklinikum Tuebingen" IEC project no: 252/2013B01 and 386/2017BO1). Ethical approval for use of human samples for the current study was obtained from "Medizinische Fakultät Ethik-Kommission, Rheinische Friedrich-Wilhelms-Universität, project no. 236/18 (2018)". All samples are listed in Supplementary Table 1.

**Molecular cloning**. For lentiviral constructs Tau-GFP /-Fusion Red (FR), human four repeat-domain (4 R) Tau (amino acids 243 to 375) with mutations P301L and V337M was fused to GFP or FR (Evrogen) with an 18-amino acid flexible linker (EFCSRRYRGPGIHRSPTA), as described previously (thereafter termed Tau-GFP, Tau-FR)[85]. Coding regions were cloned into the lentiviral vector pRRL.sin.PPT.hCMV.Wpre via BamHI and SalI[32]. For the generation of non-fusogenic VSV-G mutant W72A[35] and VSV-G K47A not binding to LDL receptors[37], mutations were inserted into the open reading frame of vesicular stomatitis virus glycoprotein VSV-G in pMD2.VSV-G using the Q5 SDM kit (NEB) (supplementary Table 2). For lentiviral expression, the VSV-G coding region was cloned into lentiviral vector pRRL.sin.PPT.hCMV.Wpre via BamHI and XhoI. Myc epitope-tagged SARS-CoV-2 (2019-nCoV) spike S cDNA (VG40589-CM) and Flag epitope-tagged human ACE2 cDNA (HG10108-NF) plasmids were purchased from Sino Biological.

**Cell lines**. N2a, L929, CAD5 and HEK 293 T cells were cultured in Opti-MEM (Gibco) supplemented with glutamine, 10 % (v/v) fetal bovine serum (FCS) (PAN-Biotech GmbH) and antibiotics. Human primary astrocytes (ScienCell) were cultivated as recommended by ScienCell. Vero cells were purchased from CLS (Cell lines service) and cultivated as recommended. All cells were incubated at 37 °C and 5% $CO_2$. The total numbers of viable cells and the viability of cells were determined using the Vi-VELL$^{TM}$XR Cell Viability Analyzer (Beckman Coulter). Transfections of cells were performed either with Lipofectamine 2000 or TransIT-2020/X2 (Mirus) reagents as recommended by the manufacturers.

**Production and transduction with lentiviral particles**. HEK293T cells were cotransfected with plasmids pRSV-Rev, pMD2.VSV-G, pMDl.g/pRRE, and pRRl.sin.PPT.hCMV.Wpre containing Tau-GFP/FR using Lipofectamine 2000. Supernatants were harvested and concentrated by the PEG method according to published protocols[86]. Cell lines and primary neurons were transduced with lentivirus, and stable cell clones expressing Tau-GFP/FR were selected following limiting dilution cloning[31].

**Extracellular vesicle isolation**. To prepare exosome-depleted medium, FCS was ultracentrifuged at 100,000× g, 4 °C for 20 h. Medium supplemented with the exosome-depleted FCS and antibiotics was subsequently filtered through a 0.22 and a 0.1 μM filter-sterilization device (Millipore). For EV isolation, 2–4 × 10$^6$ cells were seeded in a T175 flask in 35 ml exosome-depleted medium to be confluent after 3 days. For pseudotyping EV, cells were transfected with pcDNA3.1 or pMD2.VSV-G using Lipofectamine 2000. 5 h post transfection, the medium was switched to medium with EV-depleted serum. EV were harvested 3-day post transfection. Cells and cell debris were pelleted by differential centrifugation (300× g, 10 min; 2000× g, 20 min; 16,000× g, 30 min). The remaining supernatant (conditioned medium) was subjected to ultracentrifugation (UC) (100,000× g, 1 h) using rotors Ti45 or SW32Ti (Beckman Coulter). The pellet was rinsed with PBS and spun again using rotor SW55Ti (100,000× g, 1 h).

**Aggregate induction assay**. Recipient cells were cultured on CellCarrier-96 or 384 black microplates (PerkinElmer) at appropriate cell numbers for 1 h, and then treated with 5–10 μl of EV. For coculture, a total of 10$^4$ cells/ per well of recipient

and donor cells was plated at different ratios depending on their population doubling times. After additional 1 day for NM expressing and 2 day for Tau expressing cultures, cells were fixed in 4 % paraformaldehyde, and nuclei were counterstained with Hoechst. Cells were imaged with the automated confocal microscope CellVoyager CV6000 (Yokogawa Inc.) using a 20 × or 40 × objective. Maximum intensity projections were generated from Z-stacks. Images from 16 fields per well were taken. On average, a total of 3–4 × 10$^3$ cells per well and at least three wells per treatment were analyzed.

**Determination of extracellular vesicles size and number**. ZetaView PMX 110-SZ-488 Nano Particle Tracking Analyzer (Particle Metrix GmbH) was used to determine the size and number of isolated extracellular vesicles. The instrument captures the movement of extracellular particles by utilizing a laser scattering microscope combined with a video camera. For each measurement the video data is calculated by the instrument and results in a velocity and size distribution of the particles. For nanoparticle tracking analysis, the Brownian motion of the vesicles from each sample was followed at 22 °C with properly adjusted equal shutter and gain. At least three individual measurements of 11 positions within the measurement cell and around 2200 traced particles in each measurement were detected for each sample.

**Sedimentation assay for Tau**. Sedimentation assay was performed as described previously[47]. Briefly, cleared cell lysate with 100 μg total protein was centrifuged at 100,000× g for 1 h. The pellet was washed with 1.5 ml PBS and centrifuged at 100,000× g for 30 min. Supernatant fractions were precipitated with 4 × methanol at −20 °C overnight, and protein was pelleted at 2120× g for 25 min at 4 °C (soluble fraction). The pellet (insoluble fraction) and 1/3 of the soluble fraction dissolved in RIPA buffer with 4% SDS were loaded for Western blot analysis.

**Pronase digestion for Tau**. Pronase digestion experiment was performed as described previously[47]. Briefly, 18 μl cleared cell lysate or brain homogenate (20–100 μg based on Tau aggregate content) were incubated with 2 μl 1 mg/ml pronase (Roche) at 37 °C for one hour. Afterwards, samples were boiled with 3 × sample buffer, and Tau was detected by Western blot as described below.

**Brain homogenate preparation and clarification**. Frozen human brain samples were homogenized in lysis buffer (for protein analysis) via Precellys® 24 (Bertin Instruments) with 1.4 mm ceramic beads at 4 °C for 4 cycles 5500 rpm 20 s. To prepare 10 % (w/v) clear brain homogenate for aggregate induction, crude homogenates were centrifuged at 872× g for 5 min at 4 °C, and then the supernatants were sonicated with 50 % power for 6 min. These homogenates were frozen at −80 °C until use. For protein analysis, cleared supernatants were prepared by centrifugation of the crude homogenates at 15,000× g for 15 min.

**Tau aggregate induction by brain homogenate and liposomes**. To induce Tau aggregation in the HEK Tau-GFP$^{sol}$ clone with brain homogenates from different tauopathy patients, cells were plated on six well plates at 1 × 10$^6$ cells/ well in 2 ml complete medium one day before. Next day, 200 μl 10% brain homogenate and 4 μl Lipofectamine 2000 were incubated for 20 min and added to recipient cells to have final 1% brain homogenate on cells. After 3 days, cells were split and further expanded for limited dilution clone selection[31].

**PK treatment for PrP$^{Sc}$**. A cell pellet collected from one well of a six well plate was lysed in 1 ml lysis buffer. 900 μl of lysates were digested with 20 μg/ ml proteinase K (PK) at 37 °C for 30 min for PrP$^{Sc}$ detection. Proteolysis was terminated by addition of 0.5 mM Pefabloc. The remaining 100 μl lysates for total PrP detection and the digested samples were precipitated with methanol and analyzed by Western blot using anti-PrP antibody 4H11[87].

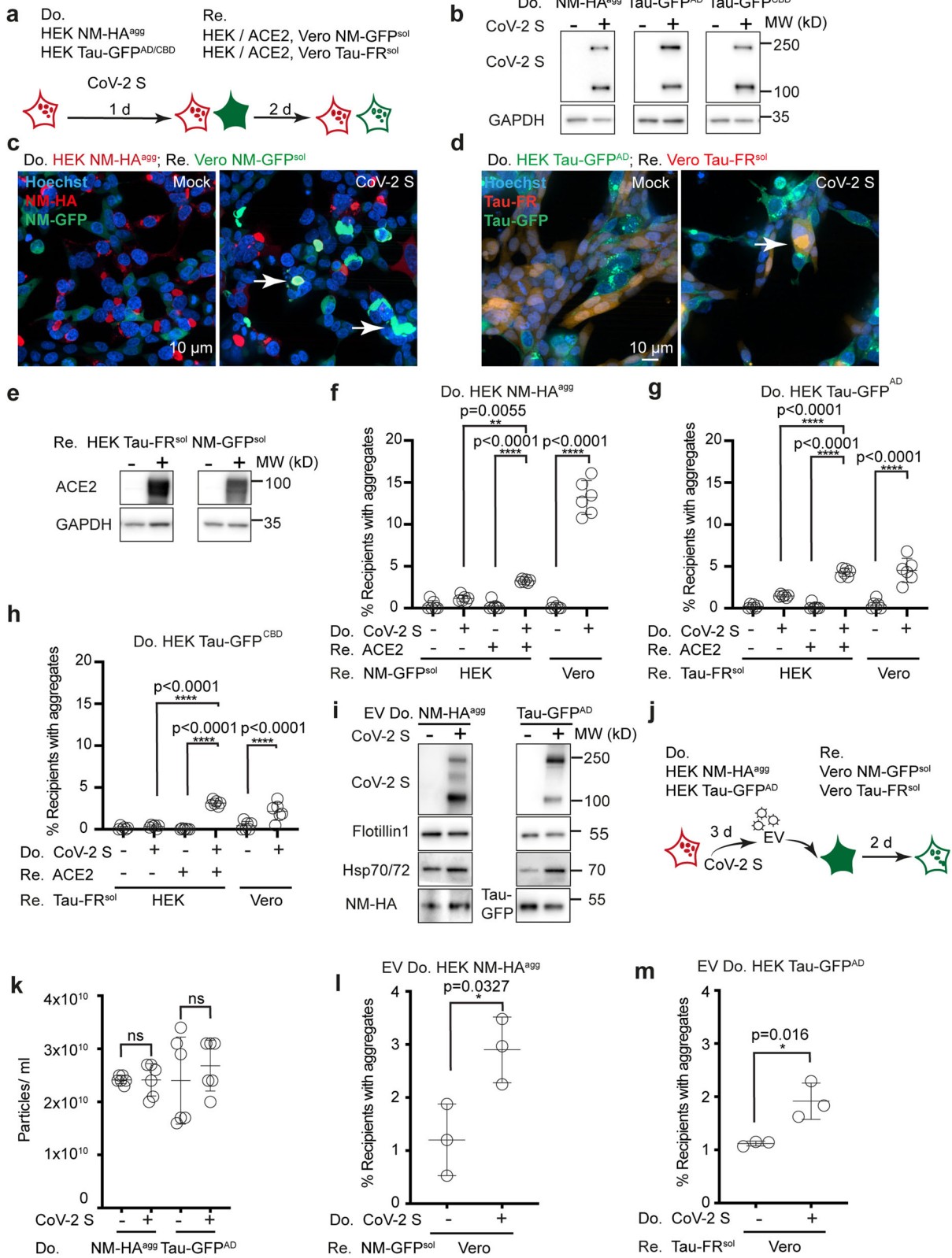

**Western blotting**. For Western blot analysis, protein concentrations were measured by Quick Start™ Bradford Protein assay (Bio-Rad) using the plate reader Fluostar Omega BMG (BMG Labtech) and the corresponding MARS Data Analysis Software (BMG Labtech). Proteins were separated on NuPAGE®Novex® 4–12 % Bis-Tris Protein Gels (Life Technologies) followed by transfer onto a PVDF membrane (GE Healthcare) in a wet blotting chamber. Western blot analysis was performed using mouse anti-LDL receptor (1:500; NovusBiologicals); mouse anti-

Alix (1:1000; BD Bioscience); rat anti-HA 3F10 (1:1000; Roche); mouse anti-GAPDH 6C5 (1:5000; Abcam); mouse anti-Hsc/Hsp70 N27F3-4 (1:1000; ENZO); mouse anti-VSV-G 5D4 (1:1000; Sigma); rat anti Sup35 M domain (1:10; hybridoma supernatant);[88] rabbit anti-Tau ab64193 (1:1000; Abcam); rabbit anti-Flotillin 1 ab133497 (1:1000; Abcam); mouse anti-SARS-CoV-2-spike S GTX632604 (1:1000; GeneTex); rabbit anti-hACE2 ab15348 (1:1000; Abcam); rabbit anti-clathrin heavy chain (1:1000; Abcam); rabbit anti-GFP (1:5000;

**Fig. 6 SARS-CoV-2 spike S protein affects spreading of proteopathic seeds. a**. HEK NM-HA$^{agg}$ or Tau-GFP$^{agg}$ cells, transfected with plasmid coding for CoV-2 spike S or Mock transfected, were cocultured with respective recipient HEK cells overexpressing/not overexpressing human ACE2. Alternatively, donors were cocultured with Vero cells stably expressing NM-GFP$^{sol}$ or Tau-FR$^{sol}$. **b** Expression of CoV-2 spike S in donor HEK cells. **c** Cocultures of transfected HEK NM-HA$^{agg}$ donors and recipient Vero NM-GFP$^{sol}$ cells. **d** Cocultures of transfected donor HEK Tau-GFP$^{AD}$ cells with recipient Vero Tau-FR$^{sol}$ cells. **e** Western blot of recipient HEK cells expressing ACE2-Flag. The blot was probed with anti-ACE2 antibodies. **f–h** Percentage of recipients with induced aggregates. **i** Presence of spike S on EV. **j** EV from donors transfected with empty or spike S plasmid were added to Vero cells. The percentage of recipients with induced aggregates was assessed 2 days later. **k** Particle numbers. **l, m** Percentage of EV exposed recipients with induced aggregates All data are shown as the means ± SD from three (**l, m**) or six (**f–h, k**) replicate cell cultures. Two (**f–h, k–m**) independent experiments were carried out with similar results. P values calculated by two-tailed unpaired Student´s t test (**k–m**) or one-way ANOVA (**f–h**). ns nonsignificant. Source data are provided as a Source Data file.

Abcam). The membrane was incubated with Pierce$^{TM}$ ECL Western Blotting Substrate (Thermo Fisher Scientific) according to the manufacturer´s recommendations and imaged with the Imaging system Fusion FX (Vilber Lourmat).

**Automated image analysis**. The image analysis was performed using the Cell-Voyager Analysis support software (CV7000 Analysis Software; Version 3.5.1.18). An image analysis routine was developed for single-cell segmentation and aggregate identification (Yokogawa Inc.) The total number of cells was determined based on the Hoechst signal, and recipient cells were detected by their GFP/ FR signal. Green aggregates were identified via morphology and intensity characteristics. The percentage of recipient cells with aggregated NM-GFP or Tau-FR/ Tau-GFP was calculated as the number of aggregate-positive cells per total recipient cells set to 100%. For spike S cocultures, false-positive induced recipient cells were detected due to the heterogeneity of GFP /FR expression of individual cells. The mean percentage of false positives determined in control recipient cells was substracted from all samples. Of note, negative values were sometimes obtained. For data presentation, the minimum range of Y axis was set to 0.

**Life cell imaging**. For life cell imaging, $2 \times 10^4$ cells/ well HEK NM-GFP$^{sol}$ cells were seeded on 384-well plates 2 h before EV addition. EV were isolated from donor HEK NM-HA$^{agg}$ cells transfected with VSV-G. Imaging was performed directly after EV addition using the CellVoyager6000. Images were recorded with a 20 min interval for up to 900 min postexposure.

**Immunofluorescence staining and confocal microscopy analysis**. Cells were fixed in 4% paraformaldehyde and permeabilized in 0.1% Triton X-100. HA staining was performed with Alexa Fluor 647-conjugated anti-HA antibody (MBL M180-A647; 1:500). For PrP$^{Sc}$ staining, proteins were denatured in 6 M guanidine hydrochloride for 10 min at RT to reduce the PrP$^C$ signal[89]. Cells were rinsed with PBS, blocked in 0.2% gelatine and incubated for 2 h with either 4H11[87] hybridoma supernatant diluted 1:10 in blocking solution or anti-VSV-G (8GF11; 1:400; Kerafast). After three washing steps with PBS, cells were incubated for 1 h with Alexa Fluor 488- conjugated secondary antibody (1:800), and nuclei were counterstained for 15 min with 4 µg/ ml Hoechst 33342 (Molecular Probes; 1:5000). 96 and 384 well plates were scanned with CellVoyager CV6000 (Yokogawa Inc.). Confocal laser-scanning microscopy was performed on a Zeiss LSM 800 laser-scanning microscope with Airyscan (Carl Zeiss) and analyzed via Zen2010 (Zen-Blue, Zeiss).

**Inhibition of endocytosis pathways**. For chemical inhibition of EV uptake and subsequent aggregate induction, HEK NM-GFP$^{sol}$ recipients were seeded on coated coverslips and 384-well plates. Twenty-four hours postseeding, cells were treated with solvent DMSO, 50 nM Bafilomycin A1, 100 µM Chloroquine, 50 µM Dynasore, or 200 nM Apilimod for 1–2 h at 37 ℃. Subsequently, VSV-G-coated EV were added per well. EV uptake was synchronized for 30 min at 4 ℃ before transferring cells back to 37 ℃. After 1 h, cells on coverslips were fixed with 2 % (vol/vol) paraformaldehyde and stained against the HA-epitope (ab9110, 1:250) and VSV-G (8G5F11, 1:400). Cells were imaged using LSM800 (Zeiss). Quantitative analysis of cells with NM-GFP aggregates was performed after 4 h and 24 h (no precooling step) using CellVoyager CV6000 (Yokogawa Inc.).

For knock-down of clathrin heavy chain (CLTC), HEK NM-GFP$^{sol}$ recipients were transfected with anti-CLTC siRNA at a final concentration of 30 nM (Hs_CLTC_10 FlexiTube siRNA; Qiagen) and control siRNA using lipofectamine RNAiMax. 48 h post transfection, cells were seeded on poly-l-lysine-coated coverslips and 384-well plates. VSV-G-coated EV were added 24 h later. For analyzing early events, EV uptake was synchronized by incubating cells for 30 min at 4 ℃ before transferring cells back to 37 ℃. Stained coverslips were imaged using (LSM800, Zeiss). For late events, cells plated on 384 well plates for 1 h were incubated with drugs for 1 h and VSV-G coated EV were added 1 h later. For quantitative analysis using CellVoyager, cells were fixed after 4 h and 24 h (no precooling step) post EV addition. Maximum intensity projections were generated from Z-stacks.

**Production of recombinant NM**. To purify recombinant NM-His, BL21 (DE3) competent *Escherichia coli* were transformed with 100 ng pET vector containing the coding sequence of NM with a C-terminal His-tag under control of the T7 promoter. Five ml of E. coli overnight cultures were inoculated into 250 ml LB media containing 100 µg/ ml ampicillin. Cultures were incubated at 37 ℃, 180 rpm (Multitron, Infors HT), until reaching an OD$_{600}$ of 0.8. NM-His expression was induced with 1 mM IPTG for 3 h at 37 ℃, 180 rpm. (10 min, 3000× $g$). Pellets from 1.5 l bacterial culture were pooled and lysed in 75 ml buffer A (8 M urea, 20 mM imidazole in phosphate buffer) for 1 h at RT. After sonication for $3 \times 10$ s at 50 % intensity, cell debris was pelleted for 20 min at 10,000× $g$ and the supernatant was sterile-filtered. NM-His was purified from the supernatant via IMAC using the ÄKTA pure protein purification system (GE Healthcare) together with a 5 ml HisTrap HP His-tag protein purification column (GE Healthcare). The supernatant was loaded onto the column initially washed with 25 ml buffer A. After rinsing with 75 ml buffer A, NM-His was eluted using a linear imidazole gradient from 10 mM to 250 mM imidazole (2–50% buffer B; 8 M urea, 500 mM Imidazole in phosphate buffer). NM-His containing fractions were pooled and concentrated to around 10% of the initial volume using Vivaspin 20 concentrator columns with a molecular cut-off of 10,000 Da. The protein was desalted using a 5 ml HiTrap Desalting column (GE Healthcare) and sterile-filtered PBS. Protein-containing fractions were pooled and frozen at −80 ℃.

**Transmission electron microscopy**. For transmission electron microscopy, 400 mesh copper grids (AGAR Scientific) were glow discharged for 30 s. MilliQ water and 1% Uranyl Acetate were filtered through a 0.22 µm filter directly before use. Isolated EV were spotted onto grids for 3 min and excess solution was blotted using Whatman paper. Grids were rinsed briefly with water and blotted again. EV were stained with 1% Uranyl acetate in water for 1 min, blotted dry, and imaged using a JEM-1400 120 kV Transmission Electron Microscope (JEOL, Japan) operated at 80 kV.

**Statistical analysis**. All analyses were performed using the Prism 6.0 (GraphPad Software v.7.0c). Statistical inter-group comparisons were performed using one-way ANOVA with a Bonferroni post-test or Student's unpaired t-test. The confidence interval in both tests was 95 %, p values smaller than 0.1 were considered significant. All experiments were performed in triplicates (EV experiments) or at least sextuplicates (coculture experiments) and repeated at least two times independently with similar results. Cocultures and EV experiments with CLTC knock-down or drug treatment were performed once at 4 h and once at 24 h. Experiments with spike S-pseudotyped EV were performed twice. Experiments with astrocyte EV were performed twice. Measurements were taken from distinct samples. At least 6000 cells were analyzed for quantitative analysis. Shown are the mean and the error bar representing the standard deviation (SD).

**Reporting Summary**. Further information on research design is available in the Nature Research Reporting Summary linked to this article.

## Data availability
Source data are provided with this paper.

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

## Acknowledgements

Funding was obtained from the Helmholtz Portfolio Wirkstoffforschung and from the Deutsche Forschungsgemeinschaft under Germany's Excellence Strategy within the framework of the Munich Cluster for Systems Neurology (EXC 2145 SyNergy– ID 390857198). The Switch Laboratory is supported by grants the Flanders institute for biotechnology (VIB, C0401), KU Leuven and its Industrieel Onderzoeksfonds (C24/17/075 to FR), the Funds for Scientific Research Flanders (FWO, equipment grants AKUL/15/34 - G0H1716N and I011620N, and project grant G045920N). We are grateful to Birgit Kurkowsky, DZNE Bonn, for technical support. We thank Corinne Lasmezas, Scripps Research Institute Florida, for providing CAD5 cells. Electron microscopy was performed using the Electron Microscopy Platform & Bio Imaging Core, VIB – KU Leuven Center for Brain & Disease Research.

## Author contributions

S.L., S.E.H. and I.V. designed research; S.L., A.H., S.E.H., A.H., S.H., L.P., K.K. and O.B. performed research; M.N. contributed human samples; S.L., S.E.H., S.F.L., S.M., P.D., J.S. and F.R. and I.V. analyzed data; and S.L. and I.V. wrote the paper.

## Funding

## Competing interests

The authors declare no competing interests.

## Additional information

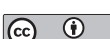

