## [Peer Review File · Nature Communications]

Reviewer comments, first round –

Reviewer #1 (Remarks to the Author):

In this manuscript from the Vorberg lab, the authors investigate whether expression of the viral surface proteins VSV-G or SARS-CoV-2 spike S influence the cell-to-cell spreading of protein aggregates in cultured cells. They find that expression of viral surface proteins on “donor” cells greatly increases the induction of protein aggregates in “recipient” cells. This is true for cells propagating aggregates of the yeast prion Sup35NM, tau, or PrPSc. The increased aggregate induction was observed both when donor cells were co-cultured with recipient cells (i.e. transfer via cell-to-cell contact) and when extracellular vesicles were isolated from donor cells and then applied to recipient cells. Induction was markedly reduced when a non-fusogenic variant of VSV-G was expressed. These results indicate that viral surface proteins can increase spreading of protein aggregates in cultured cells, raising the possibility that viral infections may contribute to the spreading of protein aggregates in neurodegenerative diseases.

This is a really exciting paper on a very hot topic. The results are striking, and the experiments are well-controlled. Even if the physiological relevance of the findings can be debated, at the very least the paradigm described in the manuscript will be a powerful tool for researchers studying the cell-to-cell propagation of protein aggregates. I do have some suggestions for improvement but am otherwise very enthusiastic about this paper.

1. I would strongly urge the authors to include a paragraph in the Discussion that outlines some of the limitations of the present study. The biggest limitation is the use of overexpressed viral surface proteins as opposed to infection of cultured cells with the actual viruses. Showing that the same phenomena occurs in virally-exposed donor cells would obviously increase the physiological relevance of the findings. Other limitations that should be discussed include: the exclusive use of immortalized cultured cells as the donor cell population (as opposed to primary or iPSC-derived neurons) and the use of tagged and/or artificial protein constructs (for instance, the repeat domain of tau containing two mutations, and then fused to a fluorescent protein). There is no question that the results are exciting, but conclusions should be tempered appropriately.

2. For the assays involving Sup35NM and tau, the authors need to show that actual aggregates are produced in the recipient cells. While fluorescent foci may be suggestive of aggregates, they could represent concentration of the soluble protein within a subcellular compartment, liquid-liquid phase separation of the protein, etc. I would suggest performing FACS to isolate the recipient cells and then conducting either detergent insolubility or protease digestion assays to test for the presence of protein aggregates.

3. The experiments shown in Figures S1c-f, S2a,b,d, and S3d need to be quantified. These are really important controls, so quantification will help to support the author’s assertions.

4. Unless I missed it, the details of the VSV-G plasmid used are not included in the Methods section.

Reviewer #2 (Remarks to the Author):

In the present paper, Liu et al. show that after expression of viral glycoproteins in extracellular vesicles (EVs), the amount of EVs and their uptake is increased, enhancing the propagation of misfolded proteins contained in and transferred by these EVs. This could have consequences for the spreading of misfolded protein seeds under viral infections. This is a well-planned study that seeks to increase the knowledge on prion-like spreading mechanisms mediated by EVs. However, some points need improvement.

MAJOR POINTS

- 1) The VSV is taken up by endocytosis through clathrin-coated pits in recipient cells. It would be interesting to check by confocal microscopy or by high-resolution microscopy how these pseudotyped EVs are taken up as the mechanism could be different from the original virus, so the internalization should be checked. If it is by classical endocytosis, can the authors explain how the content of the EVs can be released to the cytosol (where it can then act as seeds for further misfolding)?
- 2) Did the authors check the amount of LDL receptors in the recipient cells? This could explain the efficiency differences between HEK and N2a as receptors cells and should be included.
- 3) Electron microscopy of the EVs pellet showing their integrity and lack of free aggregates should be shown.
- 4) Are the western blots for VSV-G in Fig 2b representative? It seems that HEK cells present more VSV-G yet -for the same amount of EVs incubated (Fig 2g)- are not more efficiently transmitting the aggregates than N2a, which express much less VSV-G. On the contrary, in Fig. 3m it seems that the efficiency of aggregate formation in recipient cells is higher for Tau-GFPAD as donor cells than for Tau-GFPCBD, which have less VSV-G. The latter would indicate that the amount of VSV-G expression correlates with efficiency of uptake. Or can it be that the type of tau aggregates also plays a role in the aggregation process independently of the amount of VSV-G expression?
- 5) Though the data generated with VSV-G is consistent and significant, the contribution of SARS-CoV-2 to EVs uptake seems rather minimal with changes ranging from 1.5% or around 0.8% of aggregate increase in recipient cells when particle amounts are adjusted (Fig 5 l and m). Moreover, although there is evidence that SARS-CoV2 may enter the CNS, much more data must be gathered to pinpoint the number of viral particles in the brain of COVID-19 infected people. COVID-19-associated complications in the CNS of some patients may well be caused by secondary effects (systemic immune response, etc) rather than by presence/replication of the virus itself. Therefore, this point should be discussed and put in context in the discussion.

MINOR POINTS

1. It is not explained in the manuscript how EVs could express some viral proteins during cell infection. Are there some examples in the literature?
2. Sentences from 141 to 143 are a bit confusing. In 141 it is stated that the soluble isoform of Sup35 is functional, whereas the cross-beta sheet polymer is inactive. But then, in the mammalian cytosol it seems to be the other way around. As no explanation for this is given, this point may lead to confusion of prospective readers who are not that familiar with Sup35.
3. The Y axes of the graphs should be set to the same value in one figure to better visualize/interpret the changes.
4. In Fig. 4c, the pattern of PrP in L929 incubated with EVs expressing VSV-G is strange as the PK digested sample has a prominent higher band at 35 kDa that is not visible in the non-digested sample (total PrP). Do the authors have an explanation? –
5. In the material and methods section is missing how EVs were sonicated and for how long.
6. A reference is missing in the sentence in line 496.
7. In line 319 “LRP1” should be written in capital letters.

Reviewer #3 (Remarks to the Author):

In the manuscript entitled “Highly efficient intercellular spreading of protein misfolding mediated by viral ligand - receptor interactions”, the authors present data which link expression of viral proteins on extracellular vesicles (EV) to spreading of protein misfolding in three different types of cerebral protein misfolding diseases.

This paper is interesting and the topic discussed is relevant for a high level journal such as Nature communications.

However, there are a number of concerns that reduce my enthusiasm for this manuscript.

1. The authors use VSV G to stimulate uptake of EVs by recipient cells. This is of course very effective but also very artificial. The situation with Cov2-Spike and ACE2 is more realistic (but only as of end of 2019 onwards) and this certainly has not affected how dementias spread before 2019. All conclusions thus have to be seen and formulated in accordance with these facts (i.e. on page 7

the statement that: "Thus, efficient intercellular proteopathic seed transfer is strongly controlled by receptor-ligand interactions." is of course misleading as the specific receptor-ligand interactions described here do/did not occur in nature (at least not before 2019)).

2. As stated, for prions, EV spread (induced by MoMuLV) has been suggested by cell culture papers (cited in the text) yet prove of in the vivo relevance has failed in two independent studies where co-infection with F-MuLV or MoMuLV did not lead to more efficient spreading of protein misfolding (Leblanc et al. 2012; Krasemann et al. 2012), this should at least be mentioned.

3. The authors have picked up one essential factor in EV mediated spread, which is the amount of EV secreted by cells. In fact in the paper by Leblanc et al. 2006, enhanced prion spread was clearly linked to enhanced EV release by cells. This cannot be controlled in co-culture experiments. Thus the control experiment for Tau-seeding in Figure 3 is really essential. Of course this also has to be controlled for Cov2-Spike and for the prion experiments.

4. A key control experiment which has to be included to make a statement of the efficiency of spreading of protein misfolding with EV bound aggregates vs "naked" aggregates, is the sonication of EV to destroy them. Yet, one important factor has not been addressed: Could it be that sonication destroys aggregates and this then leads to less efficient spreading of protein misfolding. Since sonication is actually used to disintegrate aggregates it is likely to have an effect.

Minor points

-The statement on page 6 that: "Independent of their uptake mechanisms, EV must merge with cellular membranes to release their cargo into the cytosol" is not correct. In fact the very review which is cited states that EV contents may also reach the cytosol by leakage from lysosomal compartments.

-Unlike suggested on page 19, COVID-19 rarely presents with viral encephalitis (i.e. Solomon et al. 2020).

-It is unclear how "aggregates" are quantified in the morphological analysis in Figure 1. Judging from the show pictures in b, there is no vast difference between the GFP-signals in mock and VSV-G. Is there a biochemical way to prove enhanced aggregate formation?

Reviewer #4 (Remarks to the Author):

In their manuscript, Liu and colleagues have shown that viral glycoproteins such as VSV G and SARS-CoV-2 S strongly increase the dissemination of protein aggregates associated with neurodegenerative diseases.

This increased dissemination was shown for several proteins having prion or prion-like properties such as the NM prion domain of *Saccharomyces cerevisiae* Sup35, the aggregation competent Tau repeat-domain and PrPSc the transmissible spongiform encephalopathy agent.

This work demonstrates that receptor-ligands interactions drive direct cell-to-cell and extracellular vesicle-mediated spreading of protein misfolding and reinforces the idea that viral infections play a role in pathogenesis of neurodegenerative diseases.

The experiments have been thoroughly thought and the data are convincing. This work gives important clues on the propagation of pathogenic protein aggregates.

Nevertheless, I have some remarks that the authors should easily address.

1) The double mutant of VSV G W72A/Y73A has never been described before. Only single mutations W72A and Y73A have been described in Stanifer et al. (Ref 65 of the manuscript) and in Sun et al. (J. Biol. Chem. 2008. 283:6418–6427) (which should also be cited). Therefore, besides panel 1g, it would be good to check if the mutant is correctly transported at the cellular membranes in HEK cells (to exclude a trivial explanation of the results). This can be done using an anti-G antibody to perform immunofluorescence on non-permeabilized cells followed by flow cytometry as in Stanifer et al. or as in Ferlin et al. (J. Virol. 2014; 88: 13396-13409).

2) The authors should also use one of the mutants (either K47Q, R354A or R354Q) described in Nikolic et al. (Nat. Com. 2018. 9, Article number: 1029). These mutants do not recognize VSV receptors but have kept their fusion properties. Those mutants would be useful to investigate the role of receptor recognition in the dissemination process. By the way (line 116), the LDL receptor is not the only VSV receptor: the viral receptors are members of the LDL receptor family which all have CR domains (ref 26 and Nikolic et al. mentioned above)

Minor points.

1) In the discussion, the authors might also mention the role of the envelopes of endogenous retroviruses which are still able to catalyze membrane fusion (e.g. the syncytins).

2) The description of VSV G properties and functions is a bit naïve and incomplete. The reference 35 (line 177) is not relevant (it is about rabies virus glycoprotein not VSV G). I would rather suggest to cite the recent review of Belot et al. (Structural and cellular biology of rhabdovirus entry. *Adv Virus Res.* 2019;104:147-183.)

3) In several instances (e.g. Figure 1b, 2h, 5c, 5d), the meaning of the arrows and/or arrowheads is not explained in the legend.

POINT-BY-POINT RESPONSE TO REVIEWS

“Highly efficient intercellular spreading of protein misfolding mediated by viral ligand - receptor interactions” by Liu et al.
Manuscript #NCOMMS-20-28553-T

We would like to thank the reviewers for their thoughtful and constructive input on our manuscript. We are very pleased with the reviewer comments and welcome the opportunity to submit a revised version of our work.

We have now performed a substantial number of additional experiments and

- 1) confirm that viral glycoproteins can also increase the intercellular dissemination from primary human astrocyte donors to recipients,
- 2) carefully characterized the uptake mechanism of VSV-G coated EV,
- 3) added further control experiments to demonstrate the role of VSV-G protein in protein aggregate transmission,
- 4) used TEM to demonstrate isolation of EV,
- 5) controlled for the effect of sonication on seed integrity.

All of these additional experiments support our conclusion that viral glycoproteins affect the spreading of proteopathic seeds from cell to cell. Further, we have substantially revised the text as suggested by the reviewers.

Please find enclosed a point-by-point response to the reviewer’s comments.

Reviewer #1 (Remarks to the Author):

In this manuscript from the Vorberg lab, the authors investigate whether expression of the viral surface proteins VSV-G or SARS-CoV-2 spike S influence the cell-to-cell spreading of protein aggregates in cultured cells. They find that expression of viral surface proteins on “donor” cells greatly increases the induction of protein aggregates in “recipient” cells. This is true for cells propagating aggregates of the yeast prion Sup35NM, tau, or PrPSc. The increased aggregate induction was observed both when donor cells were co-cultured with recipient cells (i.e. transfer via cell-to-cell contact) and when extracellular vesicles were isolated from donor cells and then applied to recipient cells. Induction was markedly reduced when a non-fusogenic variant of VSV-G was expressed. These results indicate that viral surface proteins can increase spreading of protein aggregates in cultured cells, raising the possibility that viral infections may contribute to the spreading of protein aggregates in neurodegenerative diseases.

This is a really exciting paper on a very hot topic. The results are striking, and the experiments are well-controlled. Even if the physiological relevance of the findings can be debated, at the very least the paradigm described in the manuscript will be a powerful tool for researchers studying the cell-to-cell propagation of protein aggregates. I do have some suggestions for improvement but am otherwise very enthusiastic about this paper.

Response: We would like to thank you for your enthusiasm on our work and the constructive comments.

I would strongly urge the authors to include a paragraph in the Discussion that outlines some of the limitations of the present study. The biggest limitation is the use of overexpressed viral

surface proteins as opposed to infection of cultured cells with the actual viruses. Showing that the same phenomena occurs in virally-exposed donor cells would obviously increase the physiological relevance of the findings.

Response: We agree with the reviewer that the effect of active viral infection on protein aggregate transmission is very interesting. In fact, it has been shown in cell cultures that retroviral infection can enhance intercellular transmission of transmissible spongiform encephalopathy agents (cited in the manuscript). Importantly, cells infected with diverse viruses secrete both viral particles and EV containing viral protein and/or nucleic acids for intercellular spreading. As such, EV play an active role in viral infection. Thus, our experiments expressing only viral glycoproteins are of biological relevance. To clarify, we have now discussed the role of EV in viral infection in the discussion section, lines 359-368.

Other limitations that should be discussed include: the exclusive use of immortalized cultured cells as the donor cell population (as opposed to primary or iPSC-derived neurons).

Response: We agree with the reviewer that this is an important experiment and would like to thank you for suggesting this as discussion point. We had not performed these experiments as they require donor populations with high numbers of cells with the protein of interest in its aggregated state and further, high numbers of donor cells which express the viral glycoprotein. To accomplish this, we usually establish donor populations that have undergone at least 2 rounds of clonal selection before experiments can even be initiated with glycoproteins, a procedure that usually takes 2-4 months. In our hands, such a procedure proved toxic to iPS cells. We successfully managed to optimize our transduction and induction protocols over the last couple of months for human astrocytes and now demonstrate that a) VSV-G expressing donor astrocytes induce aggregation of NM-GFP in cocultured HEK NM-GFP cells (revised suppl. Fig. 3j-p) and b) that EV isolated from these donors induce NM-GFP aggregation in recipients (Fig. 2k-m). Our experiments now convincingly demonstrate that primary cells can serve as both donors and recipients for intercellular protein aggregate induction and that VSV-G expression in donors drastically increases this process. Please keep in mind that types of experiments that can be performed are very limited, as primary human astrocytes can be cultured for less than 2 weeks, with a maximum of two replatings and minimal/ no tolerance to replating after transduction. We normally isolate EV from approx. $5 \times 10^6 - 2 \times 10^7$ cells. We are unable to establish such vast numbers of donors from primary cells using the established protocol.

and the use of tagged and/or artificial protein constructs (for instance, the repeat domain of tau containing two mutations, and then fused to a fluorescent protein). There is no question that the results are exciting, but conclusions should be tempered appropriately.

*Response: To demonstrate that the effect of viral glycoproteins on intercellular protein aggregate transmission is independent of cargo, we used three independent protein aggregates, both tagged (NM, Tau) **and untagged** (PrP^{Sc}). Our data convincingly demonstrate that independent of tag and cargo protein aggregate, donors efficiently induce aggregation of the homotypic protein in the recipient.*

For the assays involving Sup35NM and tau, the authors need to show that actual aggregates are produced in the recipient cells. While fluorescent foci may be suggestive of aggregates, they could represent concentration of the soluble protein within a subcellular compartment, liquid-liquid phase separation of the protein, etc. I would suggest performing FACS to isolate

the recipient cells and then conducting either detergent insolubility or protease digestion assays to test for the presence of protein aggregates.

Response: We have now included additional analyses demonstrating that NM-GFP forms aggregates in recipient cells using a sedimentation assay (revised suppl. Fig. 3b) and Tau-FR aggregation in recipients using pronase digest of cell lysates (revised suppl. Fig. 4k).

The experiments shown in Figures S1c-f, S2a,b,d, and S3d need to be quantified. These are really important controls, so quantification will help to support the author's assertions.

Response: We have now quantified percentages of cells with aggregates for the above mentioned control experiments, where appropriate.

Former suppl. figures S1c-f were meant as introduction to our cell assays. We now omit this figure due to redundancy. Induction rates in our NM system are usually below 1 % (e.g. revised Fig. 1c-f). We now just mention in the text that we noticed differences in aggregate inducing activity among different cell lines and cite our previous publications (lanes 139-141).

Former suppl. figure S2a: now revised suppl. Fig. S2c. Analyses was already shown in (now) revised Fig. 1e, f.

Former suppl. figure S2b; now revised suppl. Fig. S2f. New analyses shown in revised suppl. Fig. S2g.

Former suppl. figure S2d; now revised suppl. Fig. S2d (one redundant image removed). New analyses shown in revised S2e.

Former suppl. figure S3d: now revised suppl. Fig. S4d. New analyses shown in revised suppl. Fig. S4e, f.

Unless I missed it, the details of the VSV-G plasmid used are not included in the Methods section.

Response: This information can be found in the Supplementary Materials section, lanes 49-53.

Reviewer #2 (Remarks to the Author):

In the present paper, Liu et al. show that after expression of viral glycoproteins in extracellular vesicles (EVs), the amount of EVs and their uptake is increased, enhancing the propagation of misfolded proteins contained in and transferred by these EVs. This could have consequences for the spreading of misfolded protein seeds under viral infections. This is a well-planned study that seeks to increase the knowledge on prion-like spreading mechanisms mediated by EVs. However, some points need improvement.

MAJOR POINTS

The VSV is taken up by endocytosis through clathrin-coated pits in recipient cells. It would be interesting to check by confocal microscopy or by high-resolution microscopy how these pseudotyped EVs are taken up as the mechanism could be different from the original virus, so the internalization should be checked. If it is by classical endocytosis, can the authors explain how the content of the EVs can be released to the cytosol (where it can then act as seeds for further misfolding)?

Response: We have now included a substantial data set, demonstrating that the preferential uptake mechanism of VSV-G coated EV is via clathrin-coated pits. First, we have assessed kinetics of VSV-G EV mediated aggregate induction in HEK NM-GFP^{sol} cells (revised Fig. 3a and suppl. movie). By confocal microscopy, we have observed that VSV-G EV tend to cluster on the cell surface once clathrin-heavy chain (CLTC) expression is silenced in recipient HEK NM-GFP^{sol} cells (revised Fig. 3b-e). Further, we have quantified productive uptake (resulting in NM-GFP aggregation) 4 h and 24 h post EV exposure under CLTC knock-down (revised Fig. 3f-h) or in the presence of endocytosis inhibitors (revised Fig. 3i-k). Our results demonstrate impairment of aggregate induction when clathrin-mediated endocytosis is blocked. Interestingly, this effect is less pronounced after 24 h, arguing that VSV-G EV can also enter by alternative routes.

Did the authors check the amount of LDL receptors in the recipient cells? This could explain the efficiency differences between HEK and N2a as receptors cells and should be included.

Response: The reviewer picks up on the interesting fact that aggregate induction by coculture was most efficient when recipient cells were HEK cells rather than N2a cells (former Fig. 1c-f). One possible explanation for this finding is that HEK cells express more LDL receptor, which constitutes the main (but not exclusive) receptor of VSV-G. We have now compared LDL receptor levels in HEK and N2a recipient populations (revised suppl. Fig. S2b). Cell populations express comparable levels of receptors, but differ slightly in receptor processing. Further, we have compared NM-GFP expression levels in HEK and N2a cells (revised suppl. Fig. S2a). Both cell lines express substantial amounts of NM-GFP substrate. HEK cells further generate an additional NM-GFP containing cleavage product, which appears to be dimerizing. It is unclear if substrate or receptor processing affects aggregate induction. However, as multiple other cellular events (e.g. endosomal escape) can affect EV-mediated aggregate induction efficiency, with our current experimental evidence we prefer not to speculate on potential mechanisms.

Electron microscopy of the EVs pellet showing their integrity and lack of free aggregates should be shown.

Response: We have now included TEM images of EV isolated from conditioned medium of HEK NM-HA^{agg} (revised Fig. 2b) and Tau-GFP^{agg} (revised suppl. Fig. S4i) expressing or not VSV-G, demonstrating cup-shaped EV characteristic for TEM. For isolation of EV, we have used a standard EV isolation method “differential centrifugation coupled with ultracentrifugation” for EV¹. Please note that this method, as other purification methods, also concentrates proteins such as lipoproteins and serum albumin present in the conditioned medium^{2,3}. We had and have therefore performed extensive control experiments demonstrating that intact EV pseudotyped with VSV-G for attachment and fusion are required for efficient aggregate induction. These control experiments include: 1) VSV-G mutants with impaired receptor binding or fusogenic activity (revised Fig. 1g-i, revised suppl. Fig. S3h, i); 2) exposure of cells to sonication-disrupted and intact EV (revised Fig. 2d, suppl. Fig. S3c) and 3) a demonstration that VSV-G provided in trans does not restore the seeding capacity of non-VSV-G coated EV (revised suppl. Fig. S3f, g). Combined, these experiments demonstrate that intact EV decorated with fusogenic VSV-G are the most potent aggregate inducers.

Are the western blots for VSV-G in Fig 2b representative? It seems that HEK cells present

more VSV-G yet -for the same amount of EVs incubated (Fig 2g)- are not more efficiently transmitting the aggregates than N2a, which express much less VSV-G. On the contrary, in Fig. 3m it seems that the efficiency of aggregate formation in recipient cells is higher for Tau-GFPAD as donor cells than for Tau-GFPCBD, which have less VSV-G. The latter would indicate that the amount of VSV-G expression correlates with efficiency of uptake. Or can it be that the type of tau aggregates also plays a role in the aggregation process independently of the amount of VSV-G expression?

Response: The reviewer is right that the total expression levels of VSV-G do not necessarily correlate with the induction efficiency in recipient cells (former Fig. 2b). The Western blots are representative. Generally, HEK cells are transfected more efficiently than N2a cells. The reviewer suggests that the seed type transmitted could potentially influence aggregate induction in recipients. Indeed, we have previously demonstrated that N2a cell clones produce EV with differently sized NM aggregates that differ in their seeding activity in recipient cells^{4,5}. We have now discussed that clonal differences, transgene expression levels and/or protein aggregate conformations can also influence intercellular aggregate induction in the discussion section (lanes 329-333).

Though the data generated with VSV-G is consistent and significant, the contribution of SARS-CoV-2 to EVs uptake seems rather minimal with changes ranging from 1.5 % or around 0.8 % of aggregate increase in recipient cells when particle amounts are adjusted (Fig 5 l and m).

Response: We agree that the difference in aggregate induction is striking when comparing VSV or Cov-2 glycoprotein expression. Please note that also titers of VSV-G and CoV-2 spike S pseudotyped HIV or MLV differ approx. 100 x fold⁶. Similar differences might be seen for pseudotyped EV. In vivo, Cov-2 is clearly a highly infectious viral pathogen. We have used VSV-G as proof-of-principle that viral glycoproteins can affect EV-mediated aggregate spreading. Although less efficiently, spike S can do so, too. We have now noted this in the discussion section (lanes 326-328).

Moreover, although there is evidence that SARS-CoV2 may enter the CNS, much more data must be gathered to pinpoint the number of viral particles in the brain of COVID-19 infected people. COVID-19-associated complications in the CNS of some patients may well be caused by secondary effects (systemic immune response, etc) rather than by presence/replication of the virus itself. Therefore, this point should be discussed and put in context in the discussion.

Response: We agree with the reviewer that the processes of CoV-2 mediated CNS pathology are likely multifactorial and have included a section discussing this point in the discussion section (lanes 382-390). We would like to stress that we do not postulate this is the most relevant mechanism leading to CNS complications in Neuro-SARS. Rather, we used spike S as ONE example of a viral glycoprotein related to a human virus. We now discuss the role of viruses in neurodegenerative diseases in more detail (lanes 341-350).

MINOR POINTS

It is not explained in the manuscript how EVs could express some viral proteins during cell infection. Are there some examples in the literature?

Response: Indeed, there are numerous examples that viruses exploit cellular communication routes such as tunneling nanotubes, cell contacts or EV for modulating the immune response

or for direct viral spreading. We have included a section on this in the discussion section (lanes 359-368).

Sentences from 141 to 143 are a bit confusing. In 141 it is stated that the soluble isoform of Sup35 is functional, whereas the cross-beta sheet polymer is inactive. But then, in the mammalian cytosol it seems to be the other way around. As no explanation for this is given, this point may lead to confusion of prospective readers who are not that familiar with Sup35.

*Response: We apologize for the confusion. With the statement that NM is non-functional in its soluble state in mammalian cells, we referred to the fact that the prion domain does not fulfill any function in the mammalian cells. We have now rephrased the sentences and explained in more detail the role of the NM domain in prion formation of Sup35 in *S. cerevisiae* and its use as a model protein in mammalian cells (lanes 129-132).*

The Y axes of the graphs should be set to the same value in one figure to better visualize/interpret the changes.

Response: The reviewer refers to former Figures 1c-f; 3e, j; 5f-h. We have now adjusted the Y axes to the same value for comparison (revised Figs. 1c-f; 4e, j; 6f-h). For clarity, we have normalized data to controls in the case of comparisons of the effect of VSV-G mutants or treatments on induced cells with aggregates (revised Figs. 1i, 3g, h, j, k; suppl. Fig. S3c, d, g, -i). In case data were normalized, we have noted this on the Y axes.

In Fig. 4c, the pattern of PrP in L929 incubated with EVs expressing VSV-G is strange as the PK digested sample has a prominent higher band at 35 kDa that is not visible in the non-digested sample (total PrP). Do the authors have an explanation?

Response: The cellular prion protein has two N-linked glycosylation sites with attached complex sugars and runs as un-glycosylated, mono-glycosylated and di-glycosylated bands on Western blots. Total PrP refers to both cellular PrP^C (proteinase K sensitive) and disease-associated PrP^{Sc} (partially proteinase K resistant). PrP^{Sc} refers to the same sample after proteinase K treatment- in this sample, PrP^C is totally degraded, while only the aminoterminal of PrP^{Sc} is degraded, leading to faster migration. In general, the glycosylation patterns of PrP^C and PrP^{Sc} differ⁷. Please note that we have loaded 9 x more PrP^{Sc} sample than PrP^C sample (see Methods section). For clarification, we have included information of PrP glycosylation in the Figure legend, as well as on loading of the two samples (lanes 813-816).

5. In the material and methods section is missing how EVs were sonicated and for how long.

Response: This information can be found in revised Figure legend Fig. 2d (lane 749) and Fig. 4j (lanes 802).

6. A reference is missing in the sentence in line 496.

Response: The reference has been added (suppl. Material, lane 138).

7. In line 319 “LRP1” should be written in capital letters.

Response: This has been corrected (lane 315).

Reviewer #3 (Remarks to the Author):

In the manuscript entitled “Highly efficient intercellular spreading of protein misfolding mediated by viral ligand - receptor interactions”, the authors present data which link expression of viral proteins on extracellular vesicles (EV) to spreading of protein misfolding in three different types of cerebral protein misfolding diseases.

This paper is interesting and the topic discussed is relevant for a high level journal such as Nature communications.

However, there are a number of concerns that reduce my enthusiasm for this manuscript.

Response: Thank you very much for your positive evaluation.

The authors use VSV G to stimulate uptake of EVs by recipient cells. This is of course very effective but also very artificial. The situation with Cov2-Spike and ACE2 is more realistic (but only as of end of 2019 onwards) and this certainly has not affected how dementias spread before 2019.

All conclusions thus have to be seen and formulated in accordance with these facts (i.e. on page 7 the statement that: “Thus, efficient intercellular proteopathic seed transfer is strongly controlled by receptor-ligand interactions.” is of course misleading as the specific receptor-ligand interactions described here do/did not occur in nature (at least not before 2019)).

Response: See also response to reviewer 2. We agree that VSV does not cause human viral infections, although rare cases occur. CoV-2, however, is a human pathogen and cases of Neuro-SARS are rising. We have now discussed potential causes of pathologic changes more carefully (lanes 382-387).

We used VSV-G and spike S as examples of viral glycoproteins and demonstrate strong increases in intercellular aggregate induction. We have now rephrased the sentence to “Thus, efficient intercellular proteopathic seed transfer CAN BE strongly controlled by receptor-ligand interactions”(lane 116).

As stated, for prions, EV spread (induced by MoMuLV) has been suggested by cell culture papers (cited in the text) yet prove of in the vivo relevance has failed in two independent studies where co-infection with F-MuLV or MoMuLV did not lead to more efficient spreading of protein misfolding (Leblanc et al. 2012; Krasemann et al. 2012), this should at least be mentioned.

Response: We thank the reviewer for pointing out the in vivo data in coinfection of mice with MuLV retrovirus and mouse-adapted scrapie, demonstrating that intraperitoneal coinfection left incubation times unaffected but partially modified brain pathology. As stated by the authors of both studies, the reason for the moderate effect of MuLV on prion infection could be the divergent cell tropism. We have now included a short section on these findings (lanes 356-358).

3. The authors have picked up one essential factor in EV mediated spread, which is the amount of EV secreted by cells. In fact in the paper by Leblanc et al. 2006, enhanced prion spread was clearly linked to enhanced EV release by cells. This cannot be controlled in co-culture experiments. Thus the control experiment for Tau-seeding in Figure 3 is really

essential. Of course this also has to be controlled for Cov2-Spike and for the prion experiments.

Response: We have now included an additional experiment in which we induce 22L prion infection using particle number adjusted EV (revised Fig. 5h, i). Again, we see a significant increase in prion infection when EV were decorated with VSV-G.

For the experiments using spike S glycoprotein, we have not observed significant differences in the EV numbers secreted by donor HEK NM-HA^{agg} or Tau-GFP^{agg} cells (former Fig. 6k, revised Fig. 6k). Induction rates were higher when recipients were exposed to spike S decorated EV (revised Fig. 6l, m). Thus, when exposed to comparable particle numbers, spike S coated EV exhibit a higher seeding capacity, independent of the transmitted seed.

A key control experiment which has to be included to make a statement of the efficiency of spreading of protein misfolding with EV bound aggregates vs “naked” aggregates, is the sonication of EV to destroy them. Yet, one important factor has not been addressed: Could it be that sonication destroys aggregates and this then leads to less efficient spreading of protein misfolding. Since sonication is actually used to disintegrate aggregates it is likely to have an effect.

Response: We agree with the reviewer that this is an important question and have thus performed additional experiments to control for this. First, we have performed a sedimentation assay of NM-HA^{agg} from EV fractions (revised suppl. Fig. S3e), demonstrating that NM-HA from sonicated EV is still present in the pellet fraction.

A similar experiment was performed on EV containing Tau-GFP, demonstrating that Tau-GFP remained aggregated after sonication (revised suppl. Fig. S4j).

Further, we have used the same sonication protocol for EV disruption and to fragment recombinant NM fibrils (revised suppl. Fig. S3c, d). We show that sonicated NM fibrils exhibit strongly increased seeding efficiency in recipient cells, while sonication of NM-HA aggregate bearing EV abrogates the seeding activity.

Combined with newly added experiments providing VSV-G in trans (revised suppl. Fig. S3f, g) and with our demonstration EV decorated with a fusion-incompetent VS-G mutant are unable to seed NM-GFP aggregation (revised suppl. Fig. S3h, i), our results clearly argue that intact, VSV-G coated EV are responsible for the drastic aggregate induction in recipients.

Minor points

-The statement on page 6 that: “Independent of their uptake mechanisms, EV must merge with cellular membranes to release their cargo into the cytosol” is not correct. In fact the very review which is cited states that EV contents may also reach the cytosol by leakage from lysosomal compartments.

Response: We have now rephrased the sentence to “For cargo to be released into the cytosol, EV USUALLY merge with cellular membranes” (lane 92).

Unlike suggested on page 19, COVID-19 rarely presents with viral encephalitis (i.e. Solomon et al. 2020).

Response: We have now rephrased the sentence and cite a review article that summarizes rare events in Covid-19 infections (lanes 382-387).

It is unclear how “aggregates” are quantified in the morphological analysis in Figure 1.

Judging from the show pictures in b, there is no vast difference between the GFP-signals in mock and VSV-G. Is there a biochemical way to prove enhanced aggregate formation?

Response: We apologize for presenting a poor example of our CellVoyager images in Fig. 1. For this cell-based assay, cells are plated on 384 well plates and images are subsequently taken automatically in 16 random fields. Cells from all 16 fields are used for analyses. We have now exchanged random images from former Fig. 1b with images that include more cells with aggregates (revised Fig. 1b). To illustrate the automated imaging and analysis procedure, we have now included a new supplementary figure (revised Fig. S1). Approximately $3\text{-}4 \times 10^3$ cells per well are analyzed, which is well above the 100-200 cells analyzed using conventional non-automated analyses. Our image analysis routine has been trained on our data set. We have now also included analyses on detection rates with donors expressing soluble protein in revised figures S2e and suppl. Figure S4e. For biochemical assays, we have now performed a sedimentation assay, demonstrating increased pelleted NM-GFP upon exposure to VSV-G coated EV (revised suppl. Fig. S3b). We have further included a pronase digest of cell lysates from Tau-FR cells exposed to Tau-GFP^{agg} containing EV, demonstrating pronase-resistant Tau-FR in cells induced with VSV-G decorated EV (revised Fig. S4k).

Reviewer #4 (Remarks to the Author):

Cell surface expression VSVG fusion mutants: The double mutant of VSV G W72A/Y73A has never been described before. Only single mutations W72A and Y73A have been described in Stanifer et al. (Ref 65 of the manuscript) and in Sun et al. (J. Biol. Chem. 2008. 283:6418–6427) (which should also be cited).

Therefore, besides panel 1g, it would be good to check if the mutant is correctly transported at the cellular membranes in HEK cells (to exclude a trivial explanation of the results). This can be done using an anti-G antibody to perform immunofluorescence on non-permeabilized cells followed by flow cytometry as in Stanifer et al. or as in Ferlin et al. (J. Virol. 2014; 88: 13396-13409).

Response: We have now also cited Sun et al. (lane 161). We replaced the double mutant with the published single mutant W72A and demonstrate its correct localization on the cell surface using the suggested antibody (revised Fig. 1g). Additionally, we now also tested the effect of this mutant on EV-mediated aggregate induction, again demonstrating that fusogenic activity is required for aggregate induction (revised suppl. Fig. S3h, i).

VSVG mutants not binding to LDL receptor: The authors should also use one of the mutants (either K47Q, R354A or R354Q) described in Nikolic et al. (Nat. Com. 2018. 9, Article number: 1029). These mutants do not recognize VSV receptors but have kept their fusion properties. Those mutants would be useful to investigate the role of receptor recognition in the dissemination process.

Response: We now included mutant K47A⁸ in our analyses. Interestingly, this mutant only partially impaired aggregate induction in cocultures (revised Fig. 1g-i). This is in line with published data, demonstrating that cell-cell contact and fusion can still occur even though receptor binding is impaired⁸. Further, we have tested this mutant on its effect on aggregate induction 4 and 24 h post EV addition (revised suppl. Fig. S3, h, i). Again, we observed reduced but not abolished aggregate induction. This suggests that LDL receptor family members represent attachment receptors. However, that a fusion-competent VSV-G unable to

bind to LRP1 receptor family members still increases aggregate induction supports our finding that VSV-G EV are preferentially taken up by clathrin-mediated endocytosis, but can also use alternative routes (revised suppl. Fig. S3h-i, compare to revised Fig. 3).

References

1. Thery, C., Amigorena, S., Raposo, G. & Clayton, A. Isolation and characterization of exosomes from cell culture supernatants and biological fluids. *Curr Protoc Cell Biol* **Chapter 3**, Unit 3 22 (2006).
2. Lobb, R.J. *et al.* Optimized exosome isolation protocol for cell culture supernatant and human plasma. *J Extracell Vesicles* **4**, 27031 (2015).
3. Webber, J. & Clayton, A. How pure are your vesicles? *J Extracell Vesicles* **2** (2013).
4. Liu, S., Hossinger, A., Hofmann, J.P., Denner, P. & Vorberg, I.M. Horizontal Transmission of Cytosolic Sup35 Prions by Extracellular Vesicles. *MBio* **7** (2016).
5. Liu, S., Hossinger, A., Gobbels, S. & Vorberg, I.M. Prions on the run: How extracellular vesicles serve as delivery vehicles for self-templating protein aggregates. *Prion* **11**, 98-112 (2017).
6. Crawford, K.H.D. *et al.* Protocol and Reagents for Pseudotyping Lentiviral Particles with SARS-CoV-2 Spike Protein for Neutralization Assays. *Viruses* **12** (2020).
7. Karapetyan, Y.E. *et al.* Prion strain discrimination based on rapid in vivo amplification and analysis by the cell panel assay. *PLoS One* **4**, e5730 (2009).
8. Nikolic, J. *et al.* Structural basis for the recognition of LDL-receptor family members by VSV glycoprotein. *Nat Commun* **9**, 1029 (2018).

Reviewer comments, second round –

Reviewer #1 (Remarks to the Author):

The authors have adequately addressed my comments.

Reviewer #2 (Remarks to the Author):

The authors have fulfilled all my requirements. I am pleased that the manuscript has now very much improved and, in my opinion, suitable for publication.
Just a very minor thing: in line 349 there is the acronym ND, but it is not defined before (or at least I could not find it). I suppose it is neurodegenerative diseases. Would be good if the authors could add (ND) after neurodegenerative diseases already in the introduction, line 49.

Reviewer #3 (Remarks to the Author):

The authors have adequately addressed the points I raised in my review. By addressing these points and the issues raised by other reviewers, the manuscript is in a very good shape now. The authors may consider updating the statements and the references concerning COVID-19. By now large neuropathology-autopsy series have been published and we have a really good idea of what is going on in the brain and viral encephalitis (ref. 82 Moriguchi et al.) is not a part of it. Also, our knowledge of SARS-CoV2 neuroinvasion is more complete now and it seems that entry via the vasculature and not the olfactory systems (Butowt et al. Acata NP 2021: The olfactory nerve is not a likely route to brain infection in COVID-19: a critical review of data from humans and animal models) is the most likely entry point.

Reviewer #4 (Remarks to the Author):

In their revised manuscript, Liu and colleagues have taken into consideration my previous remarks.
Particularly, they have included VSV G mutant K47A in their analyses.
I have also the feeling that they have answered the remarks of the other reviewers.
Overall, the amount of data presented in this work is impressive.
Therefore, I consider that the manuscript, that brings novel results on a hot topic, can now be published.

POINT-BY-POINT RESPONSE TO REVIEWS

“Highly efficient intercellular spreading of protein misfolding mediated by viral ligand - receptor interactions” by Liu et al.
Manuscript #NCOMMS-20-28553-T

We would like to thank the reviewers for their thoughtful and constructive input on our manuscript. We are very pleased with the reviewer comments and welcome the opportunity to submit a revised version of our work.

We have now performed a substantial number of additional experiments and

- 1) confirm that viral glycoproteins can also increase the intercellular dissemination from primary human astrocyte donors to recipients,
- 2) carefully characterized the uptake mechanism of VSV-G coated EV,
- 3) added further control experiments to demonstrate the role of VSV-G protein in protein aggregate transmission,
- 4) used TEM to demonstrate isolation of EV,
- 5) controlled for the effect of sonication on seed integrity.

All of these additional experiments support our conclusion that viral glycoproteins affect the spreading of proteopathic seeds from cell to cell. Further, we have substantially revised the text as suggested by the reviewers.

Please find enclosed a point-by-point response to the reviewer’s comments.

Reviewer #1 (Remarks to the Author):

In this manuscript from the Vorberg lab, the authors investigate whether expression of the viral surface proteins VSV-G or SARS-CoV-2 spike S influence the cell-to-cell spreading of protein aggregates in cultured cells. They find that expression of viral surface proteins on “donor” cells greatly increases the induction of protein aggregates in “recipient” cells. This is true for cells propagating aggregates of the yeast prion Sup35NM, tau, or PrPSc. The increased aggregate induction was observed both when donor cells were co-cultured with recipient cells (i.e. transfer via cell-to-cell contact) and when extracellular vesicles were isolated from donor cells and then applied to recipient cells. Induction was markedly reduced when a non-fusogenic variant of VSV-G was expressed. These results indicate that viral surface proteins can increase spreading of protein aggregates in cultured cells, raising the possibility that viral infections may contribute to the spreading of protein aggregates in neurodegenerative diseases.

This is a really exciting paper on a very hot topic. The results are striking, and the experiments are well-controlled. Even if the physiological relevance of the findings can be debated, at the very least the paradigm described in the manuscript will be a powerful tool for researchers studying the cell-to-cell propagation of protein aggregates. I do have some suggestions for improvement but am otherwise very enthusiastic about this paper.

Response: We would like to thank you for your enthusiasm on our work and the constructive comments.

I would strongly urge the authors to include a paragraph in the Discussion that outlines some of the limitations of the present study. The biggest limitation is the use of overexpressed viral

surface proteins as opposed to infection of cultured cells with the actual viruses. Showing that the same phenomena occurs in virally-exposed donor cells would obviously increase the physiological relevance of the findings.

Response: We agree with the reviewer that the effect of active viral infection on protein aggregate transmission is very interesting. In fact, it has been shown in cell cultures that retroviral infection can enhance intercellular transmission of transmissible spongiform encephalopathy agents (cited in the manuscript). Importantly, cells infected with diverse viruses secrete both viral particles and EV containing viral protein and/or nucleic acids for intercellular spreading. As such, EV play an active role in viral infection. Thus, our experiments expressing only viral glycoproteins are of biological relevance. To clarify, we have now discussed the role of EV in viral infection in the discussion section, lines 359-368.

Other limitations that should be discussed include: the exclusive use of immortalized cultured cells as the donor cell population (as opposed to primary or iPSC-derived neurons).

Response: We agree with the reviewer that this is an important experiment and would like to thank you for suggesting this as discussion point. We had not performed these experiments as they require donor populations with high numbers of cells with the protein of interest in its aggregated state and further, high numbers of donor cells which express the viral glycoprotein. To accomplish this, we usually establish donor populations that have undergone at least 2 rounds of clonal selection before experiments can even be initiated with glycoproteins, a procedure that usually takes 2-4 months. In our hands, such a procedure proved toxic to iPS cells. We successfully managed to optimize our transduction and induction protocols over the last couple of months for human astrocytes and now demonstrate that a) VSV-G expressing donor astrocytes induce aggregation of NM-GFP in cocultured HEK NM-GFP cells (revised suppl. Fig. 3j-p) and b) that EV isolated from these donors induce NM-GFP aggregation in recipients (Fig. 2k-m). Our experiments now convincingly demonstrate that primary cells can serve as both donors and recipients for intercellular protein aggregate induction and that VSV-G expression in donors drastically increases this process. Please keep in mind that types of experiments that can be performed are very limited, as primary human astrocytes can be cultured for less than 2 weeks, with a maximum of two replatings and minimal/ no tolerance to replating after transduction. We normally isolate EV from approx. $5 \times 10^6 - 2 \times 10^7$ cells. We are unable to establish such vast numbers of donors from primary cells using the established protocol.

and the use of tagged and/or artificial protein constructs (for instance, the repeat domain of tau containing two mutations, and then fused to a fluorescent protein). There is no question that the results are exciting, but conclusions should be tempered appropriately.

*Response: To demonstrate that the effect of viral glycoproteins on intercellular protein aggregate transmission is independent of cargo, we used three independent protein aggregates, both tagged (NM, Tau) **and untagged** (PrP^{Sc}). Our data convincingly demonstrate that independent of tag and cargo protein aggregate, donors efficiently induce aggregation of the homotypic protein in the recipient.*

For the assays involving Sup35NM and tau, the authors need to show that actual aggregates are produced in the recipient cells. While fluorescent foci may be suggestive of aggregates, they could represent concentration of the soluble protein within a subcellular compartment, liquid-liquid phase separation of the protein, etc. I would suggest performing FACS to isolate

the recipient cells and then conducting either detergent insolubility or protease digestion assays to test for the presence of protein aggregates.

Response: We have now included additional analyses demonstrating that NM-GFP forms aggregates in recipient cells using a sedimentation assay (revised suppl. Fig. 3b) and Tau-FR aggregation in recipients using pronase digest of cell lysates (revised suppl. Fig. 4k).

The experiments shown in Figures S1c-f, S2a,b,d, and S3d need to be quantified. These are really important controls, so quantification will help to support the author's assertions.

Response: We have now quantified percentages of cells with aggregates for the above mentioned control experiments, where appropriate.

Former suppl. figures S1c-f were meant as introduction to our cell assays. We now omit this figure due to redundancy. Induction rates in our NM system are usually below 1 % (e.g. revised Fig. 1c-f). We now just mention in the text that we noticed differences in aggregate inducing activity among different cell lines and cite our previous publications (lanes 139-141).

Former suppl. figure S2a: now revised suppl. Fig. S2c. Analyses was already shown in (now) revised Fig. 1e, f.

Former suppl. figure S2b; now revised suppl. Fig. S2f. New analyses shown in revised suppl. Fig. S2g.

Former suppl. figure S2d; now revised suppl. Fig. S2d (one redundant image removed). New analyses shown in revised S2e.

Former suppl. figure S3d: now revised suppl. Fig. S4d. New analyses shown in revised suppl. Fig. S4e, f.

Unless I missed it, the details of the VSV-G plasmid used are not included in the Methods section.

Response: This information can be found in the Supplementary Materials section, lanes 49-53.

Reviewer #2 (Remarks to the Author):

In the present paper, Liu et al. show that after expression of viral glycoproteins in extracellular vesicles (EVs), the amount of EVs and their uptake is increased, enhancing the propagation of misfolded proteins contained in and transferred by these EVs. This could have consequences for the spreading of misfolded protein seeds under viral infections. This is a well-planned study that seeks to increase the knowledge on prion-like spreading mechanisms mediated by EVs. However, some points need improvement.

MAJOR POINTS

The VSV is taken up by endocytosis through clathrin-coated pits in recipient cells. It would be interesting to check by confocal microscopy or by high-resolution microscopy how these pseudotyped EVs are taken up as the mechanism could be different from the original virus, so the internalization should be checked. If it is by classical endocytosis, can the authors explain how the content of the EVs can be released to the cytosol (where it can then act as seeds for further misfolding)?

Response: We have now included a substantial data set, demonstrating that the preferential uptake mechanism of VSV-G coated EV is via clathrin-coated pits. First, we have assessed kinetics of VSV-G EV mediated aggregate induction in HEK NM-GFP^{sol} cells (revised Fig. 3a and suppl. movie). By confocal microscopy, we have observed that VSV-G EV tend to cluster on the cell surface once clathrin-heavy chain (CLTC) expression is silenced in recipient HEK NM-GFP^{sol} cells (revised Fig. 3b-e). Further, we have quantified productive uptake (resulting in NM-GFP aggregation) 4 h and 24 h post EV exposure under CLTC knock-down (revised Fig. 3f-h) or in the presence of endocytosis inhibitors (revised Fig. 3i-k). Our results demonstrate impairment of aggregate induction when clathrin-mediated endocytosis is blocked. Interestingly, this effect is less pronounced after 24 h, arguing that VSV-G EV can also enter by alternative routes.

Did the authors check the amount of LDL receptors in the recipient cells? This could explain the efficiency differences between HEK and N2a as receptors cells and should be included.

Response: The reviewer picks up on the interesting fact that aggregate induction by coculture was most efficient when recipient cells were HEK cells rather than N2a cells (former Fig. 1c-f). One possible explanation for this finding is that HEK cells express more LDL receptor, which constitutes the main (but not exclusive) receptor of VSV-G. We have now compared LDL receptor levels in HEK and N2a recipient populations (revised suppl. Fig. S2b). Cell populations express comparable levels of receptors, but differ slightly in receptor processing. Further, we have compared NM-GFP expression levels in HEK and N2a cells (revised suppl. Fig. S2a). Both cell lines express substantial amounts of NM-GFP substrate. HEK cells further generate an additional NM-GFP containing cleavage product, which appears to be dimerizing. It is unclear if substrate or receptor processing affects aggregate induction. However, as multiple other cellular events (e.g. endosomal escape) can affect EV-mediated aggregate induction efficiency, with our current experimental evidence we prefer not to speculate on potential mechanisms.

Electron microscopy of the EVs pellet showing their integrity and lack of free aggregates should be shown.

Response: We have now included TEM images of EV isolated from conditioned medium of HEK NM-HA^{agg} (revised Fig. 2b) and Tau-GFP^{agg} (revised suppl. Fig. S4i) expressing or not VSV-G, demonstrating cup-shaped EV characteristic for TEM. For isolation of EV, we have used a standard EV isolation method “differential centrifugation coupled with ultracentrifugation” for EV¹. Please note that this method, as other purification methods, also concentrates proteins such as lipoproteins and serum albumin present in the conditioned medium^{2,3}. We had and have therefore performed extensive control experiments demonstrating that intact EV pseudotyped with VSV-G for attachment and fusion are required for efficient aggregate induction. These control experiments include: 1) VSV-G mutants with impaired receptor binding or fusogenic activity (revised Fig. 1g-i, revised suppl. Fig. S3h, i); 2) exposure of cells to sonication-disrupted and intact EV (revised Fig. 2d, suppl. Fig. S3c) and 3) a demonstration that VSV-G provided in trans does not restore the seeding capacity of non-VSV-G coated EV (revised suppl. Fig. S3f, g). Combined, these experiments demonstrate that intact EV decorated with fusogenic VSV-G are the most potent aggregate inducers.

Are the western blots for VSV-G in Fig 2b representative? It seems that HEK cells present

more VSV-G yet -for the same amount of EVs incubated (Fig 2g)- are not more efficiently transmitting the aggregates than N2a, which express much less VSV-G. On the contrary, in Fig. 3m it seems that the efficiency of aggregate formation in recipient cells is higher for Tau-GFPAD as donor cells than for Tau-GFPCBD, which have less VSV-G. The latter would indicate that the amount of VSV-G expression correlates with efficiency of uptake. Or can it be that the type of tau aggregates also plays a role in the aggregation process independently of the amount of VSV-G expression?

Response: The reviewer is right that the total expression levels of VSV-G do not necessarily correlate with the induction efficiency in recipient cells (former Fig. 2b). The Western blots are representative. Generally, HEK cells are transfected more efficiently than N2a cells. The reviewer suggests that the seed type transmitted could potentially influence aggregate induction in recipients. Indeed, we have previously demonstrated that N2a cell clones produce EV with differently sized NM aggregates that differ in their seeding activity in recipient cells^{4,5}. We have now discussed that clonal differences, transgene expression levels and/or protein aggregate conformations can also influence intercellular aggregate induction in the discussion section (lanes 329-333).

Though the data generated with VSV-G is consistent and significant, the contribution of SARS-CoV-2 to EVs uptake seems rather minimal with changes ranging from 1.5 % or around 0.8 % of aggregate increase in recipient cells when particle amounts are adjusted (Fig 5 l and m).

Response: We agree that the difference in aggregate induction is striking when comparing VSV or Cov-2 glycoprotein expression. Please note that also titers of VSV-G and CoV-2 spike S pseudotyped HIV or MLV differ approx. 100 x fold⁶. Similar differences might be seen for pseudotyped EV. In vivo, Cov-2 is clearly a highly infectious viral pathogen. We have used VSV-G as proof-of-principle that viral glycoproteins can affect EV-mediated aggregate spreading. Although less efficiently, spike S can do so, too. We have now noted this in the discussion section (lanes 326-328).

Moreover, although there is evidence that SARS-CoV2 may enter the CNS, much more data must be gathered to pinpoint the number of viral particles in the brain of COVID-19 infected people. COVID-19-associated complications in the CNS of some patients may well be caused by secondary effects (systemic immune response, etc) rather than by presence/replication of the virus itself. Therefore, this point should be discussed and put in context in the discussion.

Response: We agree with the reviewer that the processes of CoV-2 mediated CNS pathology are likely multifactorial and have included a section discussing this point in the discussion section (lanes 382-390). We would like to stress that we do not postulate this is the most relevant mechanism leading to CNS complications in Neuro-SARS. Rather, we used spike S as ONE example of a viral glycoprotein related to a human virus. We now discuss the role of viruses in neurodegenerative diseases in more detail (lanes 341-350).

MINOR POINTS

It is not explained in the manuscript how EVs could express some viral proteins during cell infection. Are there some examples in the literature?

Response: Indeed, there are numerous examples that viruses exploit cellular communication routes such as tunneling nanotubes, cell contacts or EV for modulating the immune response

or for direct viral spreading. We have included a section on this in the discussion section (lanes 359-368).

Sentences from 141 to 143 are a bit confusing. In 141 it is stated that the soluble isoform of Sup35 is functional, whereas the cross-beta sheet polymer is inactive. But then, in the mammalian cytosol it seems to be the other way around. As no explanation for this is given, this point may lead to confusion of prospective readers who are not that familiar with Sup35.

*Response: We apologize for the confusion. With the statement that NM is non-functional in its soluble state in mammalian cells, we referred to the fact that the prion domain does not fulfill any function in the mammalian cells. We have now rephrased the sentences and explained in more detail the role of the NM domain in prion formation of Sup35 in *S. cerevisiae* and its use as a model protein in mammalian cells (lanes 129-132).*

The Y axes of the graphs should be set to the same value in one figure to better visualize/interpret the changes.

Response: The reviewer refers to former Figures 1c-f; 3e, j; 5f-h. We have now adjusted the Y axes to the same value for comparison (revised Figs. 1c-f; 4e, j; 6f-h). For clarity, we have normalized data to controls in the case of comparisons of the effect of VSV-G mutants or treatments on induced cells with aggregates (revised Figs. 1i, 3g, h, j, k; suppl. Fig. S3c, d, g, -i). In case data were normalized, we have noted this on the Y axes.

In Fig. 4c, the pattern of PrP in L929 incubated with EVs expressing VSV-G is strange as the PK digested sample has a prominent higher band at 35 kDa that is not visible in the non-digested sample (total PrP). Do the authors have an explanation?

Response: The cellular prion protein has two N-linked glycosylation sites with attached complex sugars and runs as un-glycosylated, mono-glycosylated and di-glycosylated bands on Western blots. Total PrP refers to both cellular PrP^C (proteinase K sensitive) and disease-associated PrP^{Sc} (partially proteinase K resistant). PrP^{Sc} refers to the same sample after proteinase K treatment- in this sample, PrP^C is totally degraded, while only the aminoterminal of PrP^{Sc} is degraded, leading to faster migration. In general, the glycosylation patterns of PrP^C and PrP^{Sc} differ⁷. Please note that we have loaded 9 x more PrP^{Sc} sample than PrP^C sample (see Methods section). For clarification, we have included information of PrP glycosylation in the Figure legend, as well as on loading of the two samples (lanes 813-816).

5. In the material and methods section is missing how EVs were sonicated and for how long.

Response: This information can be found in revised Figure legend Fig. 2d (lane 749) and Fig. 4j (lanes 802).

6. A reference is missing in the sentence in line 496.

Response: The reference has been added (suppl. Material, lane 138).

7. In line 319 “LRP1” should be written in capital letters.

Response: This has been corrected (lane 315).

Reviewer #3 (Remarks to the Author):

In the manuscript entitled “Highly efficient intercellular spreading of protein misfolding mediated by viral ligand - receptor interactions”, the authors present data which link expression of viral proteins on extracellular vesicles (EV) to spreading of protein misfolding in three different types of cerebral protein misfolding diseases.

This paper is interesting and the topic discussed is relevant for a high level journal such as Nature communications.

However, there are a number of concerns that reduce my enthusiasm for this manuscript.

Response: Thank you very much for your positive evaluation.

The authors use VSV G to stimulate uptake of EVs by recipient cells. This is of course very effective but also very artificial. The situation with Cov2-Spike and ACE2 is more realistic (but only as of end of 2019 onwards) and this certainly has not affected how dementias spread before 2019.

All conclusions thus have to be seen and formulated in accordance with these facts (i.e. on page 7 the statement that: “Thus, efficient intercellular proteopathic seed transfer is strongly controlled by receptor-ligand interactions.” is of course misleading as the specific receptor-ligand interactions described here do/did not occur in nature (at least not before 2019)).

Response: See also response to reviewer 2. We agree that VSV does not cause human viral infections, although rare cases occur. CoV-2, however, is a human pathogen and cases of Neuro-SARS are rising. We have now discussed potential causes of pathologic changes more carefully (lanes 382-387).

We used VSV-G and spike S as examples of viral glycoproteins and demonstrate strong increases in intercellular aggregate induction. We have now rephrased the sentence to “Thus, efficient intercellular proteopathic seed transfer CAN BE strongly controlled by receptor-ligand interactions”(lane 116).

As stated, for prions, EV spread (induced by MoMuLV) has been suggested by cell culture papers (cited in the text) yet prove of in the vivo relevance has failed in two independent studies where co-infection with F-MuLV or MoMuLV did not lead to more efficient spreading of protein misfolding (Leblanc et al. 2012; Krasemann et al. 2012), this should at least be mentioned.

Response: We thank the reviewer for pointing out the in vivo data in coinfection of mice with MuLV retrovirus and mouse-adapted scrapie, demonstrating that intraperitoneal coinfection left incubation times unaffected but partially modified brain pathology. As stated by the authors of both studies, the reason for the moderate effect of MuLV on prion infection could be the divergent cell tropism. We have now included a short section on these findings (lanes 356-358).

3. The authors have picked up one essential factor in EV mediated spread, which is the amount of EV secreted by cells. In fact in the paper by Leblanc et al. 2006, enhanced prion spread was clearly linked to enhanced EV release by cells. This cannot be controlled in co-culture experiments. Thus the control experiment for Tau-seeding in Figure 3 is really

essential. Of course this also has to be controlled for Cov2-Spike and for the prion experiments.

Response: We have now included an additional experiment in which we induce 22L prion infection using particle number adjusted EV (revised Fig. 5h, i). Again, we see a significant increase in prion infection when EV were decorated with VSV-G.

For the experiments using spike S glycoprotein, we have not observed significant differences in the EV numbers secreted by donor HEK NM-HA^{agg} or Tau-GFP^{agg} cells (former Fig. 6k, revised Fig. 6k). Induction rates were higher when recipients were exposed to spike S decorated EV (revised Fig. 6l, m). Thus, when exposed to comparable particle numbers, spike S coated EV exhibit a higher seeding capacity, independent of the transmitted seed.

A key control experiment which has to be included to make a statement of the efficiency of spreading of protein misfolding with EV bound aggregates vs “naked” aggregates, is the sonication of EV to destroy them. Yet, one important factor has not been addressed: Could it be that sonication destroys aggregates and this then leads to less efficient spreading of protein misfolding. Since sonication is actually used to disintegrate aggregates it is likely to have an effect.

Response: We agree with the reviewer that this is an important question and have thus performed additional experiments to control for this. First, we have performed a sedimentation assay of NM-HA^{agg} from EV fractions (revised suppl. Fig. S3e), demonstrating that NM-HA from sonicated EV is still present in the pellet fraction.

A similar experiment was performed on EV containing Tau-GFP, demonstrating that Tau-GFP remained aggregated after sonication (revised suppl. Fig. S4j).

Further, we have used the same sonication protocol for EV disruption and to fragment recombinant NM fibrils (revised suppl. Fig. S3c, d). We show that sonicated NM fibrils exhibit strongly increased seeding efficiency in recipient cells, while sonication of NM-HA aggregate bearing EV abrogates the seeding activity.

Combined with newly added experiments providing VSV-G in trans (revised suppl. Fig. S3f, g) and with our demonstration EV decorated with a fusion-incompetent VS-G mutant are unable to seed NM-GFP aggregation (revised suppl. Fig. S3h, i), our results clearly argue that intact, VSV-G coated EV are responsible for the drastic aggregate induction in recipients.

Minor points

-The statement on page 6 that: “Independent of their uptake mechanisms, EV must merge with cellular membranes to release their cargo into the cytosol” is not correct. In fact the very review which is cited states that EV contents may also reach the cytosol by leakage from lysosomal compartments.

Response: We have now rephrased the sentence to “For cargo to be released into the cytosol, EV USUALLY merge with cellular membranes” (lane 92).

Unlike suggested on page 19, COVID-19 rarely presents with viral encephalitis (i.e. Solomon et al. 2020).

Response: We have now rephrased the sentence and cite a review article that summarizes rare events in Covid-19 infections (lanes 382-387).

It is unclear how “aggregates” are quantified in the morphological analysis in Figure 1.

Judging from the show pictures in b, there is no vast difference between the GFP-signals in mock and VSV-G. Is there a biochemical way to prove enhanced aggregate formation?

Response: We apologize for presenting a poor example of our CellVoyager images in Fig. 1. For this cell-based assay, cells are plated on 384 well plates and images are subsequently taken automatically in 16 random fields. Cells from all 16 fields are used for analyses. We have now exchanged random images from former Fig. 1b with images that include more cells with aggregates (revised Fig. 1b). To illustrate the automated imaging and analysis procedure, we have now included a new supplementary figure (revised Fig. S1). Approximately $3\text{-}4 \times 10^3$ cells per well are analyzed, which is well above the 100-200 cells analyzed using conventional non-automated analyses. Our image analysis routine has been trained on our data set. We have now also included analyses on detection rates with donors expressing soluble protein in revised figures S2e and suppl. Figure S4e. For biochemical assays, we have now performed a sedimentation assay, demonstrating increased pelleted NM-GFP upon exposure to VSV-G coated EV (revised suppl. Fig. S3b). We have further included a pronase digest of cell lysates from Tau-FR cells exposed to Tau-GFP^{agg} containing EV, demonstrating pronase-resistant Tau-FR in cells induced with VSV-G decorated EV (revised Fig. S4k).

Reviewer #4 (Remarks to the Author):

Cell surface expression VSVG fusion mutants: The double mutant of VSV G W72A/Y73A has never been described before. Only single mutations W72A and Y73A have been described in Stanifer et al. (Ref 65 of the manuscript) and in Sun et al. (J. Biol. Chem. 2008. 283:6418–6427) (which should also be cited).

Therefore, besides panel 1g, it would be good to check if the mutant is correctly transported at the cellular membranes in HEK cells (to exclude a trivial explanation of the results). This can be done using an anti-G antibody to perform immunofluorescence on non-permeabilized cells followed by flow cytometry as in Stanifer et al. or as in Ferlin et al. (J. Virol. 2014; 88: 13396-13409).

Response: We have now also cited Sun et al. (lane I61). We replaced the double mutant with the published single mutant W72A and demonstrate its correct localization on the cell surface using the suggested antibody (revised Fig. 1g). Additionally, we now also tested the effect of this mutant on EV-mediated aggregate induction, again demonstrating that fusogenic activity is required for aggregate induction (revised suppl. Fig. S3h, i).

VSVG mutants not binding to LDL receptor: The authors should also use one of the mutants (either K47Q, R354A or R354Q) described in Nikolic et al. (Nat. Com. 2018. 9, Article number: 1029). These mutants do not recognize VSV receptors but have kept their fusion properties. Those mutants would be useful to investigate the role of receptor recognition in the dissemination process.

Response: We now included mutant K47A⁸ in our analyses. Interestingly, this mutant only partially impaired aggregate induction in cocultures (revised Fig. 1g-i). This is in line with published data, demonstrating that cell-cell contact and fusion can still occur even though receptor binding is impaired⁸. Further, we have tested this mutant on its effect on aggregate induction 4 and 24 h post EV addition (revised suppl. Fig. S3, h, i). Again, we observed reduced but not abolished aggregate induction. This suggests that LDL receptor family members represent attachment receptors. However, that a fusion-competent VSV-G unable to

bind to LRP1 receptor family members still increases aggregate induction supports our finding that VSV-G EV are preferentially taken up by clathrin-mediated endocytosis, but can also use alternative routes (revised suppl. Fig. S3h-i, compare to revised Fig. 3).

References

1. Thery, C., Amigorena, S., Raposo, G. & Clayton, A. Isolation and characterization of exosomes from cell culture supernatants and biological fluids. *Curr Protoc Cell Biol* **Chapter 3**, Unit 3 22 (2006).
2. Lobb, R.J. *et al.* Optimized exosome isolation protocol for cell culture supernatant and human plasma. *J Extracell Vesicles* **4**, 27031 (2015).
3. Webber, J. & Clayton, A. How pure are your vesicles? *J Extracell Vesicles* **2** (2013).
4. Liu, S., Hossinger, A., Hofmann, J.P., Denner, P. & Vorberg, I.M. Horizontal Transmission of Cytosolic Sup35 Prions by Extracellular Vesicles. *MBio* **7** (2016).
5. Liu, S., Hossinger, A., Gobbels, S. & Vorberg, I.M. Prions on the run: How extracellular vesicles serve as delivery vehicles for self-templating protein aggregates. *Prion* **11**, 98-112 (2017).
6. Crawford, K.H.D. *et al.* Protocol and Reagents for Pseudotyping Lentiviral Particles with SARS-CoV-2 Spike Protein for Neutralization Assays. *Viruses* **12** (2020).
7. Karapetyan, Y.E. *et al.* Prion strain discrimination based on rapid in vivo amplification and analysis by the cell panel assay. *PLoS One* **4**, e5730 (2009).
8. Nikolic, J. *et al.* Structural basis for the recognition of LDL-receptor family members by VSV glycoprotein. *Nat Commun* **9**, 1029 (2018).

Reviewer comments, third round –

Reviewer #1 (Remarks to the Author):

The authors have adequately addressed my comments.

Reviewer #2 (Remarks to the Author):

The authors have fulfilled all my requirements. I am pleased that the manuscript has now very much improved and, in my opinion, suitable for publication.
Just a very minor thing: in line 349 there is the acronym ND, but it is not defined before (or at least I could not find it). I suppose it is neurodegenerative diseases. Would be good if the authors could add (ND) after neurodegenerative diseases already in the introduction, line 49.

Reviewer #3 (Remarks to the Author):

The authors have adequately addressed the points I raised in my review. By addressing these points and the issues raised by other reviewers, the manuscript is in a very good shape now. The authors may consider updating the statements and the references concerning COVID-19. By now large neuropathology-autopsy series have been published and we have a really good idea of what is going on in the brain and viral encephalitis (ref. 82 Moriguchi et al.) is not a part of it. Also, our knowledge of SARS-CoV2 neuroinvasion is more complete now and it seems that entry via the vasculature and not the olfactory systems (Butowt et al. *Acata NP 2021: The olfactory nerve is not a likely route to brain infection in COVID-19: a critical review of data from humans and animal models*) is the most likely entry point.

Reviewer #4 (Remarks to the Author):

In their revised manuscript, Liu and colleagues have taken into consideration my previous remarks.
Particularly, they have included VSV G mutant K47A in their analyses.
I have also the feeling that they have answered the remarks of the other reviewers.
Overall, the amount of data presented in this work is impressive.
Therefore, I consider that the manuscript, that brings novel results on a hot topic, can now be published.

POINT-BY-POINT RESPONSE TO REVIEWERS

“Highly efficient intercellular spreading of protein misfolding mediated by viral ligand - receptor interactions” by Liu et al.

Manuscript #NCOMMS-20-28553-T

July 20, 2012

We would like to thank the reviewers for their thorough review of our revised manuscript and their positive feed-back.

Reviewer #1 (Remarks to the Author):

The authors have adequately addressed my comments.

Response: Thank you for your positive feed-back.

Reviewer #2 (Remarks to the Author):

The authors have fulfilled all my requirements. I am pleased that the manuscript has now very much improved and, in my opinion, suitable for publication.

Just a very minor thing: in line 349 there is the acronym ND, but it is not defined before (or at least I could not find it). I suppose it is neurodegenerative diseases. Would be good if the authors could add (ND) after neurodegenerative diseases already in the introduction, line 49.

Response: We are pleased that we have adequately addressed your concerns. We apologize and have now introduced the abbreviation “ND” on lane 49 and used this abbreviation in the manuscript.

Reviewer #3 (Remarks to the Author):

The authors have adequately addressed the points I raised in my review. By addressing these points and the issues raised by other reviewers, the manuscript is in a very good shape now.

The authors may consider updating the statements and the references concerning COVID-19. By now large neuropathology-autopsy series have been published and we have a really good idea of what is going on in the brain and viral encephalitis (ref. 82 Moriguchi et al.) is not a part of it. Also, our knowledge of SARS-CoV2 neuroinvasion is more complete now and it seems that entry via the vasculature and not the olfactory systems (Butowt et al. Acata NP 2021: The olfactory nerve is not a likely route to brain infection in COVID-19: a critical review of data from humans and animal models) is the most likely entry point.

Response: Thank you for your positive evaluation and for pointing out the interesting article on entry routes of CoV-2. We now cite this reference (reference 84) and name olfactory and/or vascular entry routes for CoV-2 invasion of the CNS (lanes 387-389). As controversy exists on the role of CoV-2 in the brain, we prefer not to put too much emphasis on this discussion. Please note that we are stating that viral encephalitis is rare, but has been reported (lanes 384-386).

Reviewer #4 (Remarks to the Author):

In their revised manuscript, Liu and colleagues have taken into consideration my previous remarks. Particularly, they have included VSV G mutant K47A in their analyses.

I have also the feeling that they have answered the remarks of the other reviewers.

Overall, the amount of data presented in this work is impressive. Therefore, I consider that the manuscript, that brings novel results on a hot topic, can now be published.

Response: Thank you very much for this positive evaluation.